# On the influence of cell shape on dynamic reaction-diffusion polarization patterns

**K. Eroumé[1], A. Vasilevich[2], S. Vermeulen[1,2], J. de Boer[2], A. Carlier[1]***

**1** MERLN Institute for Technology-Inspired Regenerative Medicine, Maastricht University, Maastricht, The Netherlands, **2** Dept. of Biomedical Engineering, Eindhoven University of Technology, Eindhoven, The Netherlands

* a.carlier@maastrichtuniversity.nl

**Data Availability Statement:** We have provided a link to the standard polarization model which is available as a public model in the VCell repository as Kerbai_PLoSone_2021_teardrop_polarization_ extended for the extended model and

## Abstract

The distribution of signaling molecules following mechanical or chemical stimulation of a cell defines cell polarization, with regions of high active Cdc42 at the front and low active Cdc42 at the rear. As reaction-diffusion phenomena between signaling molecules, such as Rho GTPases, define the gradient dynamics, we hypothesize that the cell shape influences the maintenance of the "front-to-back" cell polarization patterns. We investigated the influence of cell shape on the Cdc42 patterns using an established computational polarization model. Our simulation results showed that not only cell shape but also Cdc42 and Rho-related (in) activation parameter values affected the distribution of active Cdc42. Despite an initial Cdc42 gradient, the *in silico* results showed that the maximal Cdc42 concentration shifts in the opposite direction, a phenomenon we propose to call "reverse polarization". Additional *in silico* analyses indicated that "reverse polarization" only occurred in a particular parameter value space that resulted in a balance between inactivation and activation of Rho GTPases. Future work should focus on a mathematical description of the underpinnings of reverse polarization, in combination with experimental validation using, for example, dedicated FRET-probes to spatiotemporally track Rho GTPase patterns in migrating cells. In summary, the findings of this study enhance our understanding of the role of cell shape in intracellular signaling.

## Introduction

Cells are shaped by the actin-myosin cytoskeleton and the surrounding environment [1–3]. The interaction between actin and myosin generates contractile stresses that produce tension at the cell cortex, which defines, amongst others, the mechanical and shape properties of the cell. Rho GTPases (Cdc42, Rac, Rho) and phosphoinositides (PIP, PIP2, PIP3) are signaling molecules that regulate the reorganization of the actin cytoskeleton [3–9]. When a graded or localized signal is applied to a cell, a cell polarizes, by redistributing the activity of these signaling molecules. Typically, regions of high Cdc42 or Rac activity will mark the front of the cell while rear regions of polarized cells are characterized by high Rho activity [3–9] (Fig 1). Correct cell polarization is essential for cell motility and is, as such, observed in various

Kerbai_PLoSone_2021_teardrop_polarization_minimal for the minimal model under the user name KerbaicBITE. Model codes (extended and minimal) are provided in the supplementary (S28_Models). Details on how to run a model in VCell can be found in the quick start guide on the VCell website, https://vcell.org/support.

**Funding:** AC and JDB kindly acknowledge the Dutch province of Limburg in the LINK (FCL67723) ("Limburg INvesteert in haar Kenniseconomie") knowledge economy project. AC acknowledges a VENI grant (number 15075) from the Dutch Science Foundation (NWO). SV is supported by the European Union's Horizon 2020 Programme (H2020-MSCA-ITN-2015; Grant agreement 676338). The funders had no role in the study design, data collection and analysis, decision to publish or preparation of the manuscript.

**Competing interests:** The authors have declared that no competing interests exist.

physiological and pathological processes such as tissue development, wound healing, and cancer invasion [10–12]. Interestingly, the timescale of polarization varies greatly with cell type, ranging from 30 seconds for highly motile neutrophils to 30–50 minutes for fibroblasts [13].

By confining cells to (a)symmetric initial shapes, using micropatterned surfaces, and measuring the preferential direction of motion after release from the confinement, Jiang et al. [14] showed that cells initially patterned in asymmetric shapes (such as a teardrop and a triangle) preferentially moved towards the direction of the blunt end. Cells initially patterned in symmetric shapes (such as a square and circle) moved out of the patterns in a random fashion, suggesting that cell shape defines the distribution of active Cdc42 and Rac. Interestingly, by exploring similar shapes of confinement but with different adhesive micropatterns (e.g., a full triangle versus a V-shape), Jiang et al. [14] also showed that it was the asymmetry of the initial cell shape, rather than the asymmetry in the initial distribution of adhesions that determined the direction of cell motion. However, *how* cell shape affects the direction of cell motion is still unknown. In this study, we hypothesize that cell shape influences the reaction-diffusion phenomena between Rho GTPases and PIPs, influencing (the maintenance of) the polarization patterns and, in turn, the cell migration direction. Due to the complex interactions between Rho GTPases and the difficulty to decouple cell shape from cell polarization experimentally (as they continuously feed back on each other) [4], we aim to use computational modelling tools to analyze cell shape-dependent polarization patterns.

Several mathematical models of cell polarization have been proposed which capture various levels of interaction between GTPases and other signaling molecules [4–7, 15–24]. More specifically, model simulations involving active and inactive Cdc42, Rac, and Rho in one and two dimensions for non-motile and deforming cells are presented in [4, 5, 7, 19]. Chiou et al. [25] show that a broad class of Rho GTPase polarization models all exhibit the ability to switch between unipolar (e.g., the front of a migrating cell) or multipolar (e.g., multiple dendrites of a neuron) outcomes. They show that this switch in model behavior is primarily determined by the saturation point that sets the maximum local Rho GTPase concentration and influences the competition between local concentration peaks. Another level of complexity is introduced by the positive feedback to and from Rac and Cdc42 as well as by linking the PIs to the Rho

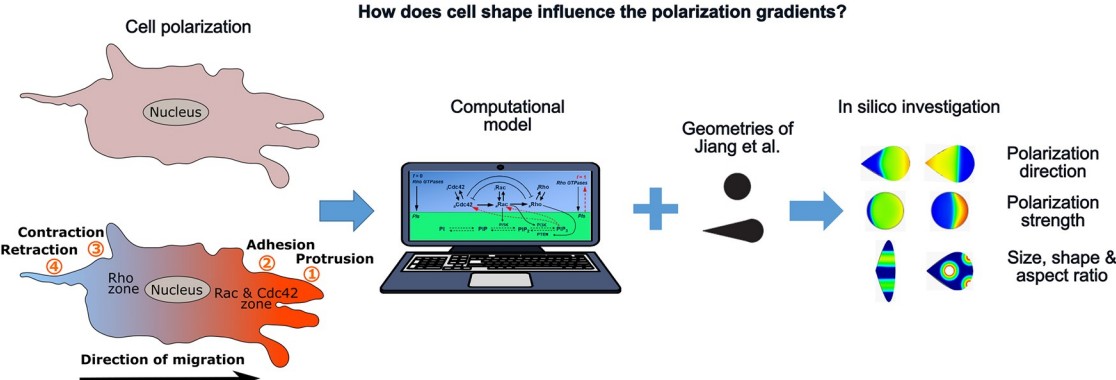

**Fig 1. Summary chart of the investigation of the influence of cell shape on cell polarization gradients.** From left to right; schematic overview of cell polarization (top) Unstimulated cell (bottom) Following stimulation (e.g. mechanical or biochemical) of a cell, active Rho GTPases redistribute such that regions of high active Rac and Cdc42 represent the front while the back is associated with regions of a high amount of active Rho as suggested by [4]. We implemented the computational model of Marée et al. [4] on the shapes used by Jiang et al. [14] to investigate the influence of cell shape on polarization, including the effect of polarization direction, polarization strength, size, shape and aspect ratio.

GTPases as discussed in [4, 19, 23, 26, 27]. Marée et al. [4] have shown that PI signaling affects the distribution of Rho GTPases, which is necessary for the cell to resolve competing cell "fronts". They also show how cell shape can influence the distribution of active Rho GTPases, albeit not systematically and rigorously in confined cells of various shapes. Narang et al. [19] and Subramanian et al. [20], (2004), have shown that chemoattractant gradient sensing is linked to the formation of a local PI peak in cells, which is in turn connected to the signaling pathways that drive actin polymerization towards lamellipodium extension. Park et al. further incorporated cell-extracellular matrix contacts into a Rac-Rho cell polarization model [28]. They demonstrated that the spatially distributed, mechanochemical feedback coupling the dynamically changing cell-extracellular matrix contacts to the activation of Rho GTPases can explain the random, persistent and oscillatory migration modes occurring in populations of melanoma cells with different degrees of aggressiveness. Bergmann et al. [29] showed that mass-conserved reaction-diffusion models of cell polarization (like the one of Otsuji et al. [20]) can be classified as active phase separation, described by the universal Cahn-Hilliard model. Recently, the polarization models were also extended to a 3D bulk-surface model [30].

Building on the acquired knowledge detailed above, focusing mostly on establishing a polarization gradient mathematically, this study aims to explore the effect of cell shape on cell polarization patterns in a rigorous and systematic way. Hereto we implement the computational model proposed by Maree et al. [4] for the cell shapes developed by Jiang et al. [14]. We then proceed to explore the influence of cell aspect ratio, PI-feedback, the direction and strength of the initial stimulus, the inactivation rate of the Rho GTPases, and strength of the negative feedback from Rho to Cdc42 on the emerging cell polarization patterns.

## Materials and methods

### Polarization model

In order to study the relationship between cell shape and cell polarization, we implemented the model developed by Marée et al. [4] for the shapes studied by Jiang et al. [14]. The model implementations were done in Virtual Cell; a computational simulation and modelling platform [31].

The computational model of Marée et al. [4] includes three modules: Rho GTPases, PIs and the cystoskeleton. As we focus on static, confined cell geometries in this study, we only model the first two modules (i.e., Rho GTPases and PIs). Fig 2 depicts the schematic summary of the signaling cascade, which is represented by a set of 9 PDEs that describe the evolution

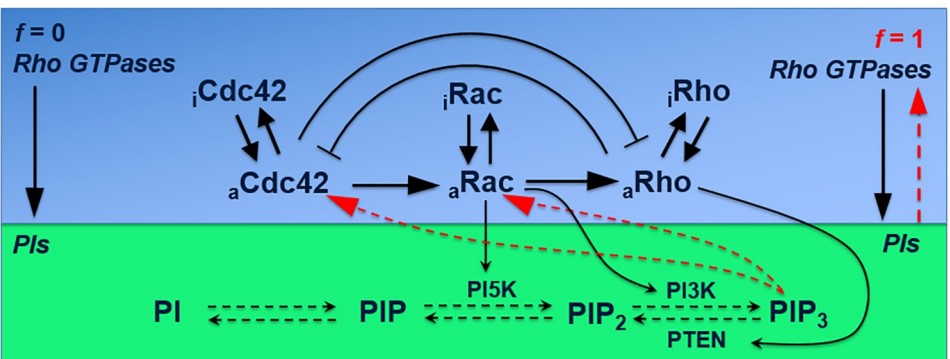

**Fig 2. Schematic overview of the Rho GTPase model adapted from [4].** The f parameter corresponds to the feedback between Rho GTPases and PIs. In this study, we set the f parameter to 0.4.

(including the kinetics, crosstalk, diffusion) of active and inactive Cdc42, Rac and Rho and membrane diffusing PIP, PIP2, PIP3. Note that the *in silico* model describes the concentrations of the active proteins as an approximation for their activity. To account for the strength of the feedback between PIs and Rho GTPases, the parameter f ($0 \leq f \leq 1$) was used; with f = 0 representing no feedback and f = 1 representing full dependence of Rho GTPase activation on PI feedback.

The set of partial differential equations was numerically solved using the fully implicit finite volume regular grid solver with a variable time step of VirtualCell. The PDE model was solved on a 2D grid with mesh size 110x110 elements, an absolute tolerance of $10^{-9}$ and a relative tolerance of $10^{-7}$. Prior to selecting the appropriate mesh size, we performed a mesh convergence analysis, details of which are found in the S1 Fig in S1 File. The Virtual Cell simulations were remotely run on the high performance computing facilities of the Center for Cell Analysis and Modelling (CCAM) of the University of Connecticut Health campus at Farmington, CT. The code is available through the VCell repository.

**Model equations.** *Rho GTPases.* The evolution of the active forms is described by Eq 1:

$$\frac{\partial G}{\partial t} = Q_G(C, R, \rho, P_3)\left(\frac{G_i}{G_{tot}}\right) - d_G G + D_m \Delta G, \tag{1}$$

$G = C, R, \rho$ represent the active forms of Cdc42, Rac and Rho respectively; $G_{tot} = C_{tot}, R_{tot}, \rho_{tot}$ represent the total concentrations of Cdc42, Rac and Rho, and $G_i = C_i, R_i, \rho_i$ the concentrations of the inactive forms of Cdc42, Rac and Rho. The inactive forms (which are located in the cytosol) diffuse much faster than the active forms (which are bound to the membrane), therefore $D_m \ll D_{mc}$. The evolution of the inactive forms is modelled as follows:

$$\frac{\partial G_i}{\partial t} = -Q_G(C, R, \rho, P_3)\left(\frac{G_i}{G_{tot}}\right) + d_G G + D_{mc} \Delta G_i. \tag{2}$$

$D_{mc}$ represents the effective cytosolic diffusion of the inactive forms of the Rho GTPases and it is approximated by $D_{mc} = f_{mem}D_m + f_{cyt}D_c$ with $f_{cyt}$ and $f_{mem}$ representing respectively the average fraction of cytosolic ($f_{cyt}$) and membrane $f_{mem}$ time spent by an inactive protein (see also Table 1).

The activation of Cdc42, Rac, Rho is given by:

$$Q_c^0 = \frac{I_c}{\left(1 + \left(\frac{\rho}{a_1}\right)^n\right)}, Q_R^0 = (I_r + \alpha C), Q_\rho = \frac{(I_p + \beta R)}{\left(1 + \left(\frac{C}{a_2}\right)^n\right)} \tag{3}$$

$I_c, I_r, I_\rho$ correspond to the baseline activation rates of Cdc42, Rac and Rho and $a_1$ and $a_2$ are the concentrations that describe the half-maximal drop of Cdc42 and Rho activation, respectively. The rate of Rac amplification by Cdc42 and the rate of Rho amplification by Rac is given by $\alpha$ and $\beta$ respectively where n is the hill coefficient.

The feedback of PIP$_3$ to Rho GTPases activation (green module in Fig 2) is captured by modifying $Q_c^0$ and $Q_R^0$ in Eq 4 as follows:

$$Q_c = \frac{I_c}{\left(1 + \left(\frac{\rho}{a_1}\right)^n\right)}\left([1-f] + f\frac{P_3}{P_{3b}}\right),$$

$$Q_R = (I_c + \alpha C)\left([1-f] + f\frac{P_3}{P_{3b}}\right) \tag{4}$$

**Table 1. Parameter values for the polarization model, same as Marée et al. [4].**

| Parameter | Definition | Value |
|---|---|---|
| $C_{tot}$, $R_{tot}$, $P_{tot}$ | total levels of Cdc42, Rac, Rho | 2.4, 7.5, 3.1 μM |
| $I_C$, $I_R$, $I_\rho$ | Cdc42, Rac, Rho activation input rates | 2.95, 0.5, 3.3 μMs$^{-1}$ |
| $a_1$ | Rho level for half-max inhibition of Cdc42 | 1.25 μM |
| $a_2$ | Cdc42 level for half-max inhibition of Rho | 1 μM |
| n | Hill coeff. of Cdc42-Rho mutual inhibition | 3 |
| f | Strength PI-feedback | 0.4 |
| α | Cdc42-dependent Rac activation rate | 4.5 s$^{-1}$ |
| β | Rac-dependent Rho activation rate | 0.3 s$^{-1}$ |
| $d_C$, $d_R$, $d_\rho$ | decay rates of activated Rho-proteins | 1 s$^{-1}$ |
| $D_m$, $D_{mc}$ | diffusion coefficient of active, inactive Cdc42, Rac, Rho | 0.1, 50 μm$^2$ s$^{-1}$ |
| $C_b$, $R_b$, $\rho_b$ | typical basal levels of active Cdc42, Rac, Rho | 1, 3, 1.25 μM |
| $I_{P1}$ | PIP$_1$ input rate | 10.5 μMs$^{-1}$ |
| $\delta_{P1}$ | PIP$_1$ decay rate | 0.21s$^{-1}$ |
| $k_{PI5K}$ | PIP$_1$ to PIP$_2$ baseline conversion rate (by PI5K) | 0.084 μM$^{-1}$ s$^{-1}$ |
| $k_{21}$ | PIP$_2$ to PIP$_1$ conversion rate | 0.014 s$^{-1}$ |
| $kP_{I3K}$ | PIP$_2$ to PIP$_3$ baseline conversion rate (by PI3K) | 0.00072 μM$^{-1}$ s$^{-1}$ |
| $kP_{TEN}$ | PIP$_3$ to PIP$_2$ baseline conversion rate (by PTEN) | 0.432 μM$^{-1}$ s$^{-1}$ |
| $D_P$ | Diffusion rate of PIP$_1$, PIP$_2$, PIP$_3$ | 5 μm$^2$ s$^{-1}$ |
| $P_{1b}$, $P_{2b}$, $P_{3b}$ | typical basal levels of PIP$_1$, PIP$_2$, PIP$_3$ | 50, 30, 0.05 μM |

PTEN, PI3K and PI5K are considered to be well-mixed.

Here the parameter $f(0 \leq f \leq 1)$ serves to tune PIP$_3$ feedback to Cdc42 and Rac activation. It is worth noting that setting f to zero implies no feedback and Eq 4 becomes Eq 3 (with the Rho activation remaining PIP independent). In the standard model, $f = 0.4$. The baseline concentration of PIP$_3$ in a resting cell is given by $P_{3b}$.

*PI dynamics.* The evolution of PIs is modelled as follows:

$$\frac{\partial P_1}{\partial t} = I_{P1} - \delta_{P1}P_1 + k_{21}P_2 - \frac{k_{p15k}}{2}\left(1 + \frac{R}{R_b}\right)P_1 + D_P\Delta P_1 \tag{5}$$

$$\frac{\partial P_2}{\partial t} = -k_{21}P_2 + \frac{k_{p15k}}{2}\left(1 + \frac{R}{R_b}\right)P_1 - \frac{k_{p13k}}{2}\left(1 + \frac{R}{R_b}\right)P_2 + \frac{k_{PTEN}}{2}\left(1 + \frac{\rho}{\rho_b}\right)P_1 + D_P\Delta P_2 \tag{6}$$

$$\frac{\partial P_3}{\partial t} = \frac{k_{p13k}}{2}\left(1 + \frac{R}{R_b}\right)P_2 - \frac{k_{PTEN}}{2}\left(1 + \frac{\rho}{\rho_b}\right)P_3 + D_P\Delta P_3 \tag{7}$$

Here the parameter $k_{p15k}$ represents the baseline conversion rate of PIP$_1$ to PIP$_2$, $k_{21}$ the baseline conversion rate of PIP$_2$ to PIP$_1$, $k_{p13k}$ the baseline conversion rate of PIP$_2$ to PIP$_3$, $k_{PTEN}$ the baseline conversion rate of PIP$_3$ to PIP$_2$, $I_{P1}$ the activation rate of PIP$_1$ by PI, $\delta_{P1}$ the decay rate of of PIP$_1$ and $D_P$ the PI diffusion coefficient. We do not keep track of the abundant unphosphorylated PI and instead assume that the conversion from PI to PIP$_1$ occurs at a constant rate, $I_{P1}$. PIP$_1$ decays back to the unphosphorylated pool at a rate $\delta_{P1}$. All the parameter values of the polarization model, same as Marée et al. [4], can be found in Table 1.

**Initial conditions.** We used the standard initial polarization method employed by Marée et al. [4], by applying a transient initial Cdc42 activation rate for 10s as a function of horizontal

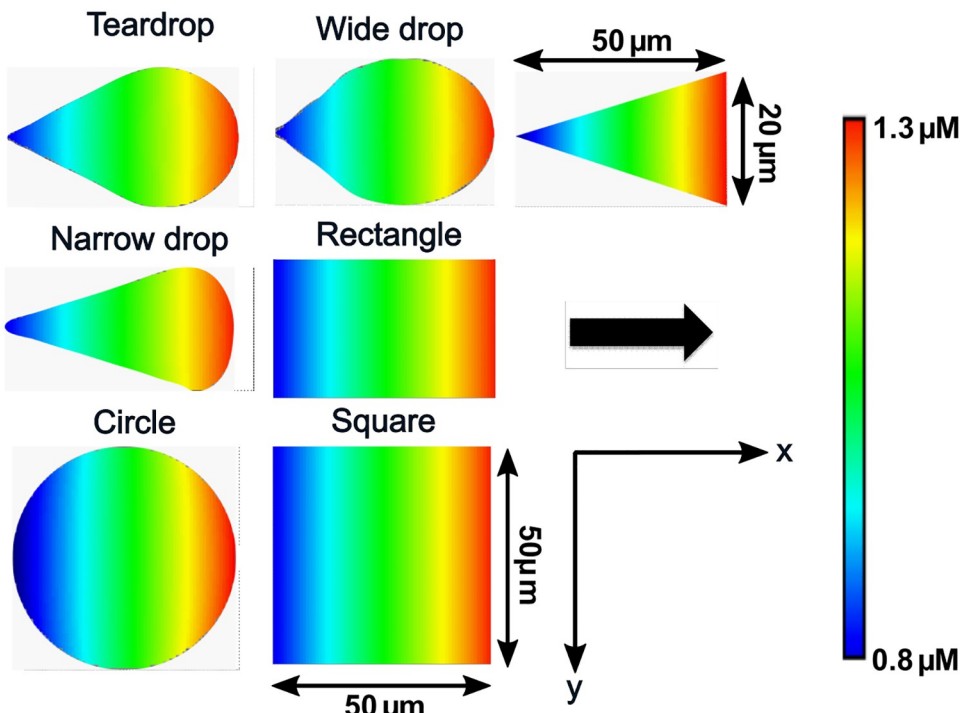

**Fig 3. Initial active Cdc42 condition and geometry for the polarization model.** Spatial distribution of active Cdc42 at 1s for all shapes (initial polarization). The black arrow indicates the initial polarization direction.

distance x (Fig 3):

$$I_c = \begin{cases} 2.6 + 0.05x, & t \leq 10s \\ 2.95, & t > 10s \end{cases} \tag{8}$$

In order to study the effect of the transient initial Cdc42 activation rate on cell polarization, we tested all combinations of three intercepts (1.6, 2.6 and 3.6) and three slopes (0.025, 0.05, 0.1). All initial conditions, implemented as uniform distributions throughout the simulation domain except for active Cdc42 are summarized in Table 2.

**Table 2. Initial concentration of each species as implemented in the VCell model.**

| Species | Initial concentration |
|---|---|
| Cdc42 inactive | 1.4 μM |
| Cdc42 active | 1 μM |
| Rac inactive | 4.5 μM |
| Rac active | 3.0 μM |
| Rho inactive | 1.85 μM |
| Rho active | 1.25 μM |
| PIP$_1$ | 55.0 μM |
| PIP$_2$ | 30.0 μM |
| PIP$_3$ | 0.05 μM |
| PI5K | 10 μM |
| PI3K | 10 μM |
| PTEN | 10 μM |

## Geometries

We distinguished between "symmetric" and "asymmetric" shapes. The symmetric shapes have more than one axis of symmetry (i.e., the circle, square and rectangle) while the asymmetric shapes only have one axis of symmetry (i.e., the teardrop, wide drop, narrow drop and triangle). The cell geometries were defined in the geometry definition menu of Virtual Cell software by using analytical expressions for the symmetric geometries, and by importing images of the asymmetric geometries (Fig 3 and S1 File).

## Boundary conditions

At the outer cell membrane a no-flux boundary condition was defined for all diffusible substances (Rho GTPases and PIs). Thus given the concentration C of any of these proteins the following equation holds: $\nabla C \cdot \hat{n} = 0$ with $\hat{n}$ corresponding to the normal unit vector at the boundary location.

### *In silico* analyses

To explore the effect of cell aspect ratio on cell polarization patterns, we varied the cellular length and width for a range of values (10 μm to 90 μm) for the teardrop, the circle (or ellipse) and the rectangle (or square). We defined the aspect ratio as the horizontal distance (cell length) divided by the vertical distance (cell width). We ran our simulations for 1000s and extended them to 3000s for those that did not converge to a single polarization pattern after 1000s. The standard polarization scheme was used (see section Initial conditions). Note that as the initial polarization gradient is dependent on the distance in the x-direction (horizontal), the initial polarization gradient will change with the aspect ratio.

For the parameter sensitivity analysis we varied the PI feedback (f = 0, 0.1, 0.8), the inactivation rate of the Rho GTPases ($d_{cdc42}$, $d_{rac}$, $d_{rho}$ = 0.01, 0.1, 1, 10 s$^{-1}$), and the strength of the negative feedback of Rho on Cdc42 ($a_1$ = 0.5, 1.25, 2 μM) for the teardrop, and circle cell shapes. We also varied the Cdc42-Rho mutual inhibition coefficient (n = 1, 4, 8), the diffusion coefficients of active ($D_{Cdc42}$, $D_{Rac}$, $D_{Rh0}$ = 0.01, 1 μm$^2$ s$^{-1}$) and inactive ($D_{Cdc42}$, $D_{Rac}$, $D_{Rh0}$ = 5, 500 μm$^2$ s$^{-1}$) Rho GTPases. Finally, we varied the diffusion coefficients of all PIs ($D_{PIP1}$, $D_{PIP}$, $D_{PIP3}$ = 5, 500 μm$^2$ s$^{-1}$). The standard polarization scheme was used. We first performed a 1D sensitivity analysis by varying the activation, inactivation and diffusion terms one-by-one. We then proceeded with a 2D sensitivity analysis by varying a1 and dCdc42 simultaneously at specific values of n (1, 4, 8).

The (threshold) maximum concentrations presented in white in the activity profiles in the results section (from Fig 4 onwards), were determined such that the maximum concentration at different time points for each species across all shapes could be visualized by the same maximum.

## Results

### Cell shape modulates the spatiotemporal Rho GTPase patterns

To examine how cell shape influences the spatial distribution of Rho GTPases, the *in silico* polarization model was applied to a variety of static cell shapes. Initially, all protein concentrations were uniform within the interior of the cell, representing an unstimulated (resting) cell. Then, the *in silico* cell was stimulated by imposing a transient (10s) spatially dependent activation of Cdc42, after which we analyzed how the local Cdc42 activation gradient as well as the other Rho GTPase and PIP concentrations evolved as a function of time and space (Fig 4). As the cell was initially polarized from left to right (black arrow in Fig 4), the maximal Cdc42

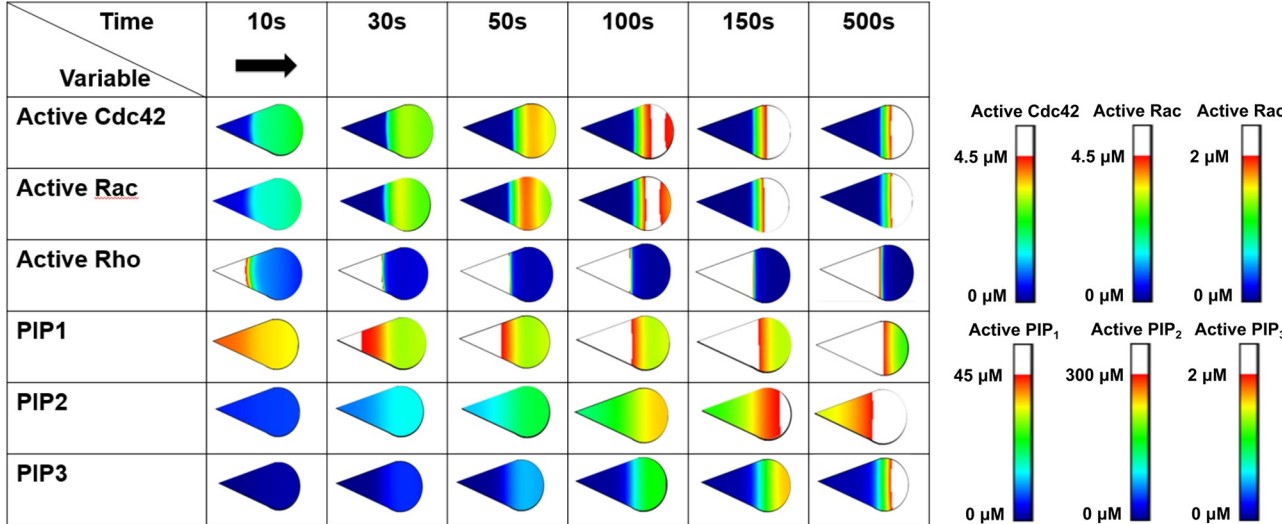

**Fig 4. Spatiotemporal evolution of the patterns of active Rho GTPases and PIPs in a teardrop-shaped cell.** Regions where the active Rho GTPase and PIP concentration is above a particular value (see scale bar) are colored white. The black arrow indicates the initial direction of polarization.

concentration was found at the front of the cell (right side in Fig 4) and the lowest Cdc42 concentration at the back (left side in Fig 4). Interestingly, despite the initial left-right polarization, the maximal Cdc42 concentration shifted in the opposite direction (Fig 4 – 100s). Depending on the shape of the cell, the duration of the initial activation and the magnitude and orientation of the initial gradient (see below), the maximal Cdc42 concentration would again move towards the right, establishing a stable left-right polarization gradient. The patterns of Rac concentration were similar to those of Cdc42; the patterns of Rho concentration did not show the shift in maximal concentration in the direction opposite the initial polarization direction (Fig 4). Note that the *in silico* model describes the concentrations of the active protein as an approximation for their activity. In the remainder of the study, we specifically chose to follow the evolution of active Cdc42 (one of the three modelled Rho GTPases) as the patterns of Rac were similar and would yield similar results (Fig 4).

To explore the influence of the direction of the initial stimulus on the Cdc42 patterns, we applied the initial stimulus in three different directions (left-right (L-R), right-left (R-L), and up-down (U-D)) to the symmetric and asymmetric cell shapes (Fig 5). Merely reversing the direction of the initial stimulus (R-L versus L-R) resulted in different Cdc42 spatial patterns. More specifically, the range of high Cdc42 concentrations (white region in Fig 5) is broader for the R-L direction compared to the L-R direction for the asymmetric shapes. Moreover, the maximum Cdc42 concentration traveled in the opposite direction of the initial polarization for all shapes but did not regain its original polarization direction for the asymmetric shapes (Fig 5, R-L). Interestingly, when the direction of the stimulus is perpendicular to the main symmetry axis of the cell shape, turning of the polarization gradient is observed for all asymmetric shapes except for the wide drop (Fig 5, U-D). Contrary to the asymmetric shapes (except for the wide drop), symmetric shapes maintain a stable polarity along the initial U-D direction after 1000s.

We also investigated the influence of the orientation of the initial stimulus on the Cdc42 patterns in relation to the duration of the initial activation and the magnitude of the initial gradient (for the teardrop and circle, Fig 6 and S5 Fig in S1 File). For the circular cell shape, the duration of the L-R or R-L stimulus did not influence whether the maximal Cdc42

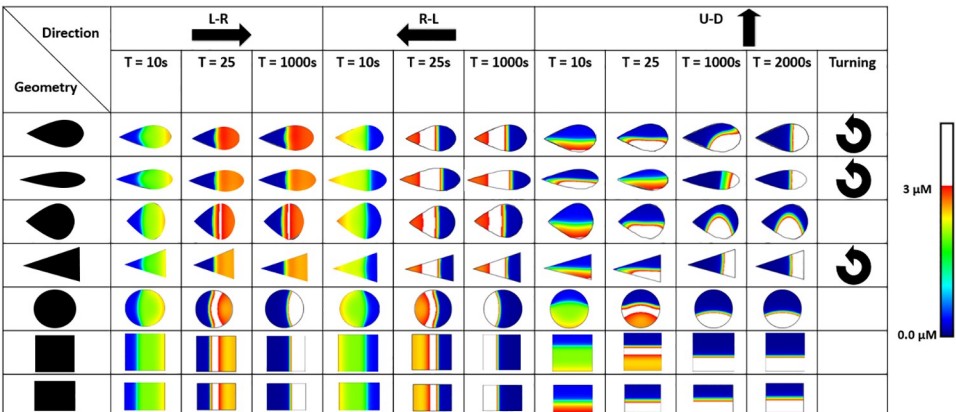

**Fig 5. Initial polarization direction influences active Cdc42 patterns.** The following initial polarization directions were tested: L-R (left-right), R-L (right-left), U-D (up-down). The turning arrow indicates which patterns turn from an up-down initial polarization pattern to a left-right. The polarization patterns represent the active Cdc42 concentration. Regions where the active Cdc42 concentration is above 3 μM are colored white.

concentration would travel in the direction opposite the initial polarization direction (Fig 6). However, for the teardrop, the Cdc42 pattern maintained the initial L-R or R-L polarization direction for longer durations of the initial stimulus (20s, sustained). Next to the duration of the initial stimulus, also the magnitude of the initial gradient affected the spatial evolution of the maximal Cdc42 concentration. Both increasing the slope or intercept of the linear gradient

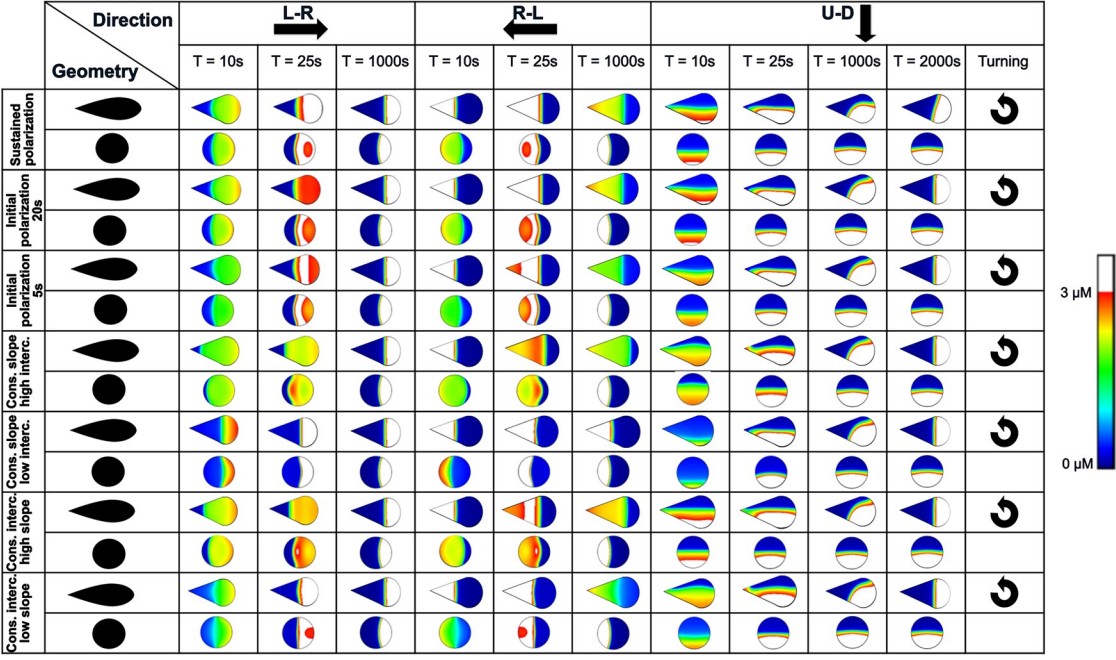

**Fig 6. Initial polarization direction in combination with the initial polarization strength and duration influences the reverse polarization pattern.** L-R (left-right), R-L (right-left), U-D (up-down) initial polarization directions. The turning arrow indicates which patterns turn from an up-down initial polarization pattern to a left-right. The polarization patterns represent the active Cdc42 concentration. Regions where the active Cdc42 concentration is above 3 μM are colored white. Constant (cons.) slope of 0.05, intercept (interc.) values: low = 1.6, high = 3. Constant intercept of 2.6, slope values: low = 0.025, high = 0.1.

resulted in a shift of the maximum in the direction opposite the initial polarization direction (termed here "reverse polarization"), independent of cell shape (Fig 6), although for the teardrop this shift was more pronounced for the R-L direction than for the L-R direction. A low intercept did not induce a shift in the maximum Cdc42 concentration (for both the circle and the teardrop) whereas a low slope did result in a shift in the maximum Cdc42 concentration opposite the original polarization direction, albeit only for the circle. Interestingly, the direction of the initial stimulus (R-L versus L-R) influenced the resulting Cdc42 patterns in the case of a teardrop shape, i.e., the shift in maximal Cdc42 concentration was more pronounced for the R-L direction than the L-R direction, independent of the duration or magnitude of the initial stimulus. When the direction of the stimulus was perpendicular to the main symmetry axis of the cell shape (U-D), turning of the polarization gradient was observed for all teardrop shapes and independent of the duration or magnitude of the initial stimulus. It is noteworthy that turning of the polarization gradient was observed for the teardrop even when a sustained U-D stimulus was applied.

We next quantified the time of onset and distance travelled by the maximal Cdc42 concentration across cell shapes for the L-R and R-L initial polarization directions (Fig 7). For the L-R

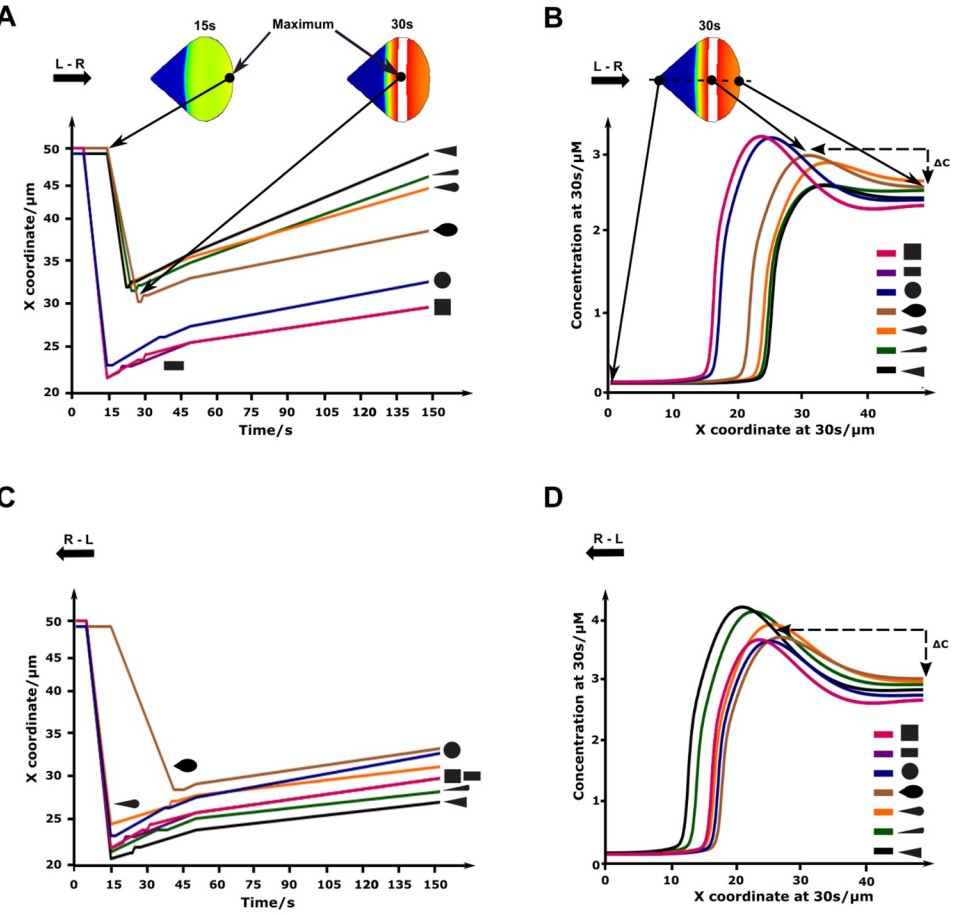

**Fig 7. Cell shape influences cell polarization patterns.** A) Temporal evolution of maximal Cdc42 for a cell stimulated in the L-R direction. B) Spatial evolution of Cdc42 concentration for a cell stimulated in the L-R direction at 30s. The concentrations are measured on the main axis, as shown in the inset. C) Temporal evolution of maximal Cdc42 for a cell stimulated in the R-L direction. D) Spatial evolution of Cdc42 concentration for a cell stimulated in the R-L direction at 30s. The concentrations are measured on the main axis, as shown in the inset, where the x-coordinate is determined from left to right.

polarization direction, the maximal Cdc42 concentration shifted earlier in the opposite direction of initial polarization in the symmetric shapes (circle, square and rectangle) compared to the asymmetric shapes (teardrop, narrow drop, triangle, wide drop) (5s versus 15s respectively, Fig 7A). The maximal Cdc42 concentration also travelled the furthest for the symmetric shapes (Fig 7A) in which it crossed the midline of the cell. At 30s the symmetric shapes reached the highest concentration (3.3 µM), which occurred furthest from the front of the cell (27 µm) (Fig 7B). Moreover, the difference between the maximal concentration and the concentration at 50 µm (the right side of the cell) is larger for the symmetric shapes. After 150s, the maximal Cdc42 concentration would again move towards the right in the asymmetric shapes, regaining the original polarization direction, which was not the case for the symmetric shapes (Fig 7A). Interestingly, for the R-L polarization, the profiles of the asymmetric and symmetric shapes do not cluster together. More specifically, the time of onset is similar in both groups, as well as the distance travelled by the maximal Cdc42 concentration (Fig 7C). At 30s (Fig 7D), the difference between the maximal concentration and the concentration at 50 µm (the right side of the cell) is now larger for the asymmetric shapes (instead of the symmetric shapes for L-R stimulation). Both asymmetric and symmetric shapes do not regain their original polarization direction after 150s. It is important to highlight that, although the domain size was constant in the direction of polarization for the L-R and R-L initial polarization experiments, it varied between cell shapes which may influence the above results regarding the shift and timing of the maximal concentration (see also Fig 8).

In summary, the orientation, magnitude and duration of the initial stimulus as well as the cell shape modulate Rho GTPase cell polarization patterns, particularly the maximal Cdc42 concentration, and its spatiotemporal location.

## Both cell shape and cell size influence cell polarization

In the previous section, differences in Cdc42 concentration patterns were found for the wide drop versus the narrow drop and the rectangle versus the square, i.e., cells with a similar shape but a different aspect ratio. Here, we wanted to explore the influence of cell aspect ratio, cell size, and cell shape on intracellular polarization patterns more rigorously and systematically. We defined the aspect ratio as the horizontal distance (cell length) divided by the vertical distance (cell width). We focused in this section on the teardrop, as an example of the asymmetric shapes and the circle and rectangle, as examples of the symmetric shapes.

Fig 8B summarizes the results across three cell shapes (i.e., rectangle, teardrop and circle). Every data point combines the results of three shapes along the two main axes, representing the cell size along these axes. The changes in polarization, which are color-coded and explained in the legend, are recorded for each of these three shapes for each axis length. After 1000s, various polarization patterns emerge for all aspect ratios and all shapes (Fig 8A, S7 and S9 Figs in S1 File).

For the teardrop, small cells (lengths ranging from 10 µm to 20 µm) polarized along the new axis as the cell size along this new axis increased (red color in Fig 8B). Further increasing the size of the cell along the new axis (beyond 30 µm), led to reverse polarization along this axis (Fig 8B), defined as shifting of the maximal Cdc42 concentration in the opposite direction of the initial polarization direction. Note the exception to this general trend, i.e., three cells with length 20 µm and widths between 40 µm– 60 µm maintain their initial polarization. Interestingly, medium-size cells (lengths ranging from 30 µm to 60 µm) maintained the direction of the initial polarization pattern, even when the cell size is increased along the new axis (Fig 8B, blue and green colors). Large cells (lengths greater than 60 µm) only show reverse polarization along the initial axis, even when the cell size along the new axis is increased.

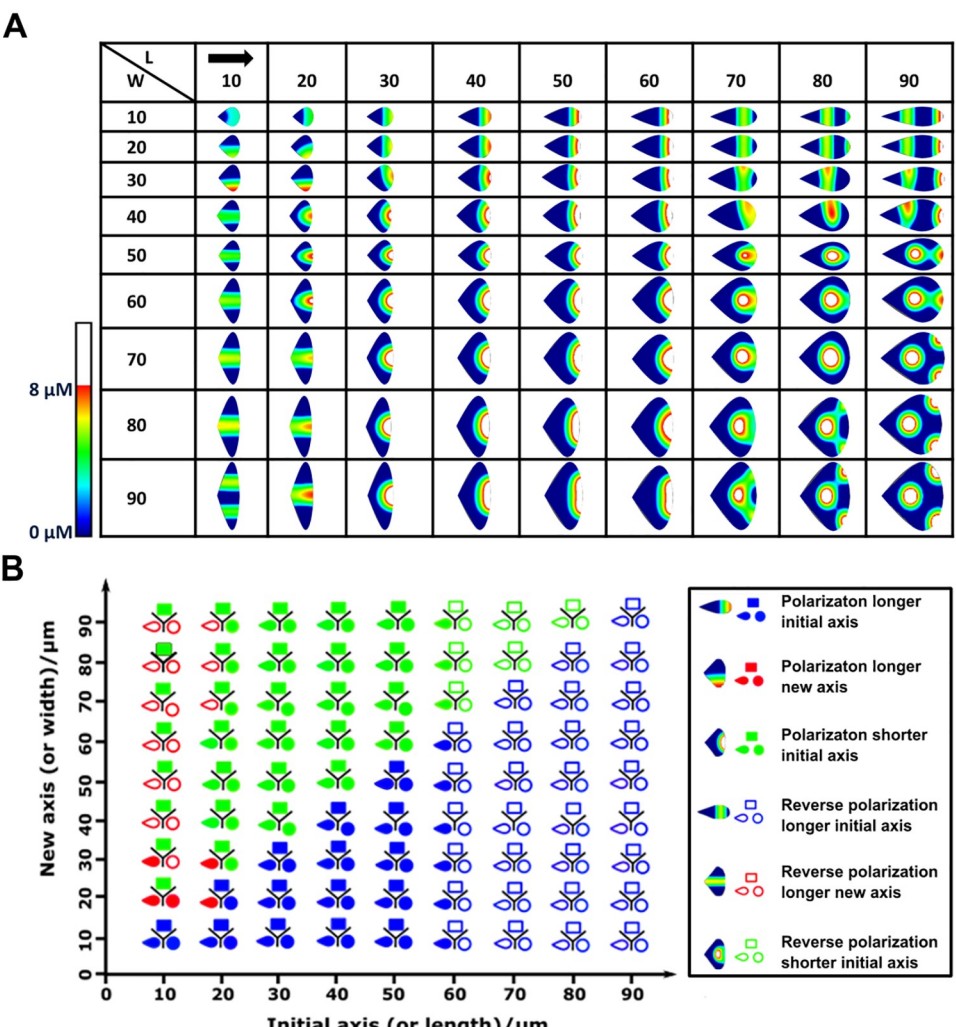

**Fig 8. Cell size and aspect ratio affect polarization patterns.** A) Polarization patterns of active Cdc42 at different aspect ratios for the teardrop. L: length, W: width. 1000s simulation time. B) Summary plot showing six main polarization patterns for the teardrop, rectangle (or square) and circle (or ellipse). The terms "shorter" and "longer" refer to the cell size along a particular (initial or new) axis with respect to the other axis. "Polarization" refers to a polarization pattern such that the maximum of Cdc42 is at one end of the cell, while the minimum is at the opposite end. "Reverse polarization" refers to the shifting of the maximum such that it is no longer situated at the one of the extremities of the cell. The "initial" axis refers to the axis along which we initially polarize the cell (L-R), while the "new" axis refers to the main axis perpendicular to the initial axis. The initial polarization direction is left-right, as indicated by the black arrow. Regions where the active Cdc42 concentration is above 8 μM are colored white.

For medium and large cells the polarization patterns of asymmetric (teardrop) and symmetric (square and circle) cells are identical, except that asymmetric shapes start to exhibit reverse polarization at larger lengths (Fig 8B). Interestingly, only for small cells the asymmetric and symmetric shapes show distinct polarization patterns depending on the cell aspect ratio. The (small) rectangular cells did not show any reverse polarization and maintained their polarization along the direction of the initial stimulus, independent of the cell size along the new axis. When the cell size increased along the new axis, the circle and teardrop first polarized along the new axis and then reverse polarized along the new (i.e., longer) axis (Fig 8B). Note that cells with a curved membrane (teardrop and circle) can turn their polarization gradient along a new axis, whereas this is not possible in rectangular cells due to the no flux boundary

conditions which ensure that the isoclines of Rho GTPases and PIs are always perpendicular to the cell membrane.

Multiple Cdc42 maxima emerged after 1000s for several teardrop and circular cells, i.e., large, stretched cells along one axis (two fronts) or both axes (three fronts) (Fig 8A). To check if these multiple maxima were stable, we ran more extended simulations (3000s) on a sample of representative cells. With the exception of aspect ratios 90x40 μm and 90x90 μm of the teardrop, all these multiple maxima converged to a single maximum at 3000s (see S6, S8 and S10 Figs in S1 File).

In conclusion, we could distinguish and summarize the various polarization patterns in six distinct patterns across all shapes (Fig 8B): polarization longer initial axis, polarization longer new axis, polarization shorter initial axis, reverse polarization longer initial axis, reverse polarization longer new axis and reverse polarization shorter initial axis. Teardrops and circles showed all six, while squares showed four types of polarization patterns (Fig 8B, S7 and S9 Figs in S1 File). These findings show that cell size and cell aspect ratio influence cell polarization, and that for very large or stretched cells multiple fronts can emerge. Interestingly, only for short cells in the direction of the initial stimulus, there are clear differences between cell shapes, and in particular, the rectangular versus the circular and teardrop-shaped cells.

### Exploring the parameter space of reverse polarization

To explore the dependence of the shifting of the maximal Cdc42 concentration in the direction opposite the initial polarization direction (termed here "reverse polarization") on the parameter values, we looked in more detail at Eq 1, which describes the spatiotemporal evolution of the Rho GTPases. The change in local Cdc42 concentration depends on an activation, inactivation, and diffusion term. We hypothesized that, depending the parameter values, the balance between activation and inactivation is different locally, moving the maximal Cdc42 concentration in the opposite direction of the initial polarization direction. More specifically, the initial polarization is maintained when this difference is positive while reverse polarization occurs when this term is negative. We first altered the parameter values of the activation, inactivation and diffusion terms in a one-at-a-time analysis (spatial results are shown in Fig 9, detailed quantifications can be found in S11–S21 Figs in S1 File).

Altering the (membrane) diffusion coefficient of the active Rho GTPases resulted in reverse polarization for both the circle and the teardrop (see Fig 9 and S14A, S14B Fig in S1 File for detailed quantifications). In contrast, reduced (cytosolic) diffusion of the inactive Rho GTPases resulted in a larger shift of the maximal Cdc42 concentration in the opposite direction of the initial polarization direction, although the concentration difference between the maximal Cdc42 concentration and the concentration at 50 μm (right-hand side of the cell) was lower than for the standard condition (Fig 9 and S14C, S14D Fig in S1 File). Similar behavior was observed when altering the diffusion coefficients of the PIPs (Fig 9 and S15 Fig in S1 File). For higher values of the inhibition coefficient n (i.e., 4–8), reverse polarization patterns were observed for both the teardrop and the circle (Fig 9 and S16 Fig in S1 File). Interestingly, for an inhibition coefficient n equal to 1 the reverse polarization pattern disappeared overall. When we varied the inactivation rate of the active Rho GTPases (Fig 9 and S16–S19 Figs in S1 File), the reverse polarization pattern disappeared for both higher (i.e., 10, see S19 Fig in S1 File) as well as lower (i.e., 0.01 and 0.1, see Fig 9, S17 and S18 Figs in S1 File) values of $d_{Cdc42}$ and $d_{rho}$. For $d_{rac}$ reverse polarization patterns could be observed independent of its parameter value (i.e., 0.01, 0.1 or 10), although the concentration difference between the maximum and the concentration at 50 μm ($\Delta C$) was small for $d_{rac} = 10$, implying a broad front of high Cdc42 concentrations (S19 Fig in S1 File).

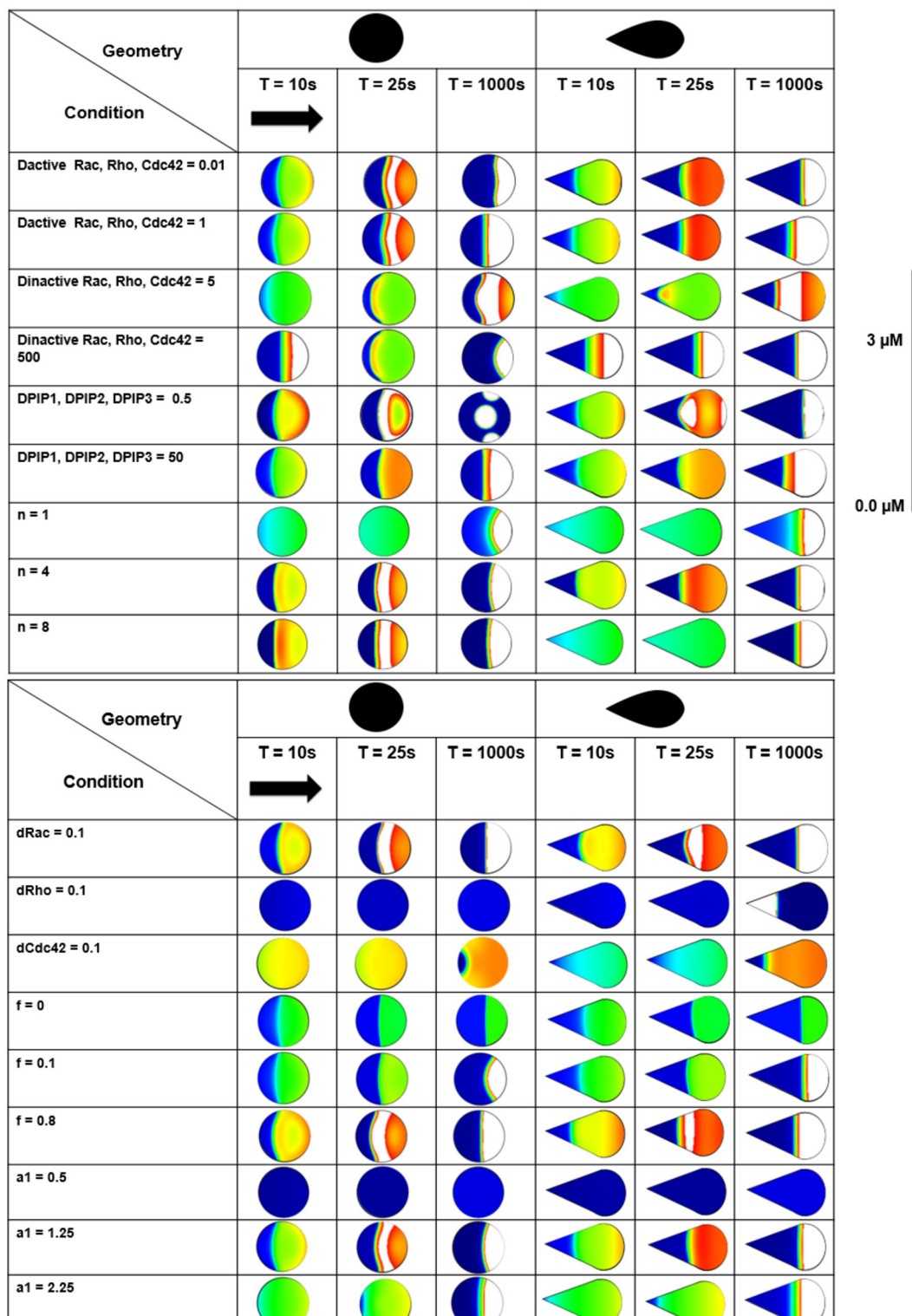

**Fig 9. 1D parameter space exploration for the circle and the teardrop.** The black arrow represents the initial polarization direction, the standard polarization scheme was used. The polarization patterns represent the active Cdc42 concentration. Regions where the active Cdc42 concentration is above 3 μM are colored white.

We also explored the influence of the f-parameter ($0 \leq f \leq 1$) which tunes the $PIP_3$ feedback to Cdc42 and Rac activation (see Methods and Fig 2). A high value of f represents a high feedback of the PI-layer to the Rho GTPase-layer, a value of f = 0 means no feedback. Fig 9 (and S20 Fig in S1 File) shows that for high PI-feedback the maximum Cdc42 concentration traveled away from its original position, resulting in reverse polarization. Although not visible in the spatial plot of Fig 9 (see S20 Fig in S1 File for details), also for low PI-feedback there is reverse polarization, albeit with a small concentration difference between the maximum and the concentration at 50 μm (ΔC). The concentration difference was also highly dependent on the cell shape and value of f (Fig 9 and S20 Fig in S1 File). More specifically, at low PI feedback there was a broad front of high Cdc42 concentrations. At high PI feedback, ΔC was larger for the symmetric shapes (S20B Fig in S1 File), consistent with the previous results (see Results section and Fig 7B and 7D).

Interestingly, a minimal Rho-Rac model (see minimal model section, including schematic overview in S22 Fig in S1 File and standard results in S23–S26 Figs in S1 File) which represents the reversible transitions between active Rac and Rho and their mutual inhibitions, neglecting the feedback of the phosphoinositide layer (similar to f = 0 in the polarization model), also displayed reverse polarization for rectangular cells (see S25 and S26 Figs in S1 File) and as such confirmed the above results. The teardrop, circle and triangle did not show any reverse polarization in the minimal model, and the reverse polarization was also not influenced by cell size and aspect ratio (S23, S24 and S26-S28 Figs in S1 File).

At intermediate levels of $a_1$ (i.e., 1.25), which represents the Rho level for half-maximal inhibition of Cdc42, reverse polarization occurred (Fig 9 and S21 Fig in S1 File). At low levels (i.e., 0.5), the Cdc42 concentration became very low and the overall gradient disappeared, whereas at high levels (i.e., 2.25) normal polarization took place (Fig 9 and S21 Fig in S1 File). Since both $d_{cdc42}$, $a_1$ and n could display reverse polarization and normal polarization patterns, dependent on their particular values in the one-at-a-time analysis, a phase-plane analysis was performed (Fig 10). By varying all three parameters in a combinatorial fashion we wanted to obtain greater insight into how each model parameter contributes to a given behavior. The average system behavior was classified as one of the following three types of phases: uniform

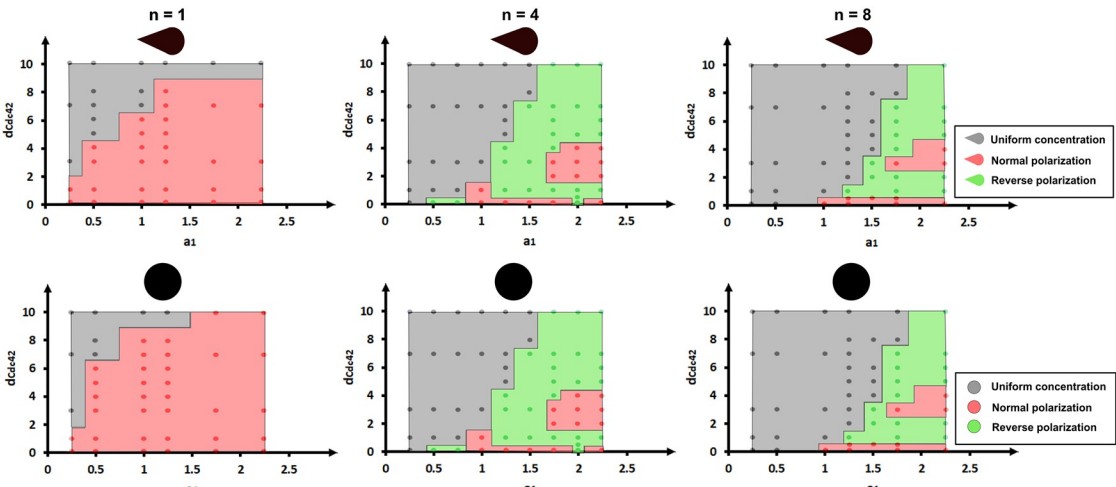

**Fig 10. Phase diagram showing how the polarization model behaves with changes in n, $a_1$ and $d_{cdc42}$ for the circle and the teardrop.** The standard polarization scheme was used. The grey phase represents a uniform concentration, the green phase represents reverse polarization and the red phase represents normal polarization. The points represent the sampling points for which simulations were run to observe the system behavior.

concentration (difference between minimum and maximum Cdc42 concentration less than 0.01 μM, black in Fig 10), normal polarization (red in Fig 10), reverse polarization (maximal Cdc42 concentration shifts away from its initial position at the right hand side of the cell, green in Fig 10). The diagrams in Fig 10 show that the phase diagrams were very similar for the teardrop and the circular cell shape. When the inhibition coefficient n was equal to 1, no reverse polarization behavior was obtained. For higher values of the inhibition coefficient (i.e., n = 4, 8), reverse polarization was observed when $a_1 > 0.5$ and $d_{cdc42} > 0.1$, although there were zones observed with normal polarization (Fig 10).

In summary, the shifting of the maximal Cdc42 concentration in the direction opposite the initial polarization direction (termed here "reverse polarization") is influenced by many parameters, including the inactivation rate of the Rho GTPases, the strength of the negative feedback, the Cdc42-Rho mutual inhibition coefficient, the diffusion coefficients of active and inactive Rho GTPases and the diffusion coefficients of PIPs.

## Discussion

Cell polarization refers to the ability of cells to form a clear defined front and back [3, 8]. It is an important process in which lipids and proteins are redistributed across the cell in response to external stimuli (mechanical or chemical), and it is involved in several processes such as cell motility, differentiation, and division [3, 8]. In this work, we studied the influence of cell shape on Cdc42 patterns *in silico*. We showed that cell shape and aspect ratio highly influence the emergent Cdc42 patterns in addition to the direction and strength of the initial polarization stimulus, the inactivation rate of the Rho GTPases, the PI feedback and strength of the negative feedback from Rho to Cdc42.

Interestingly, the *in silico* results showed that cell shape highly influences the polarization patterns emerging from an initial stimulus, and this in a non-intuitive way (see also discussion on reverse polarization below). For example, when the initial polarization direction is perpendicular to the main axis of the asymmetric shapes, the polarization gradient turns and aligns with the main axis, even when a sustained stimulus was applied (Figs 5 and 6). The tendency of the polarization wave to establish itself along the longer axis is also observed for various aspect ratios (Fig 8) and is consistent with the minimization of interface length suggested by Marée et al. [4]. More specifically, the position of the interface is determined by minimizing the length of the interface that segregates between the front and back of the cell as a result of the gradient-flow nature of the PDEs. This minimization can only occur when the polarization gradient is oriented along the longest axis. Our findings are in line with the above hypothesis, in that cells tend to polarize along the longest axis of the cell. The minimization of the interface length could also explain the resolution of multiple Cdc42 maxima into one (S6, S8, and S10 Figs in S1 File). Our observation on the progression of these peaks is similar to the observations of Mori et al. [32]. They suggest that multiple peaks are metastable and will merge into a single peak at longer simulations.

Despite an initial polarization gradient, the *in silico* results showed that the maximal Cdc42 concentration could shift in the opposite direction, a phenomenon we called "reverse polarization". To investigate the "reverse polarization" phenomenon, we explored the activation, inactivation and diffusion terms in more detail. More specifically, for medium to large cells (20 μm–90 μm), the aspect ratio influenced the Cdc42 pattern more than the (a)symmetric cell shape. For a similar aspect ratio, additional *in silico* analyses indicated that cells shift their maximal Cdc42 concentration at varying degrees depending on their shape, the aspect ratio, the direction and the strength of the initial stimulus, the inactivation rate of the Rho GTPases, the PI feedback and strength of the negative feedback from Rho to Cdc42. Moreover, by varying

the inhibition coefficient, the inactivation rate and inhibition of Cdc42 by Rho in a combinatorial fashion, we showed that when the inhibition coefficient was higher than 1 reverse polarization could occur in particular regions of the parameter space (i.e., a1 > 0.5 and dcdc42 > 0.1), although within this parameter space some zones with normal polarization were observed as well. Interestingly, also the direction of the initial stimulus (R-L versus L-R) influenced the resulting Cdc42 patterns in the case of an asymmetrical shape such as a teardrop, i.e., the shift in maximal Cdc42 concentration was more pronounced for the R-L direction than the L-R direction.

It would be interesting to perform a rigorous mathematical analysis to fully understand the (metastable) reverse polarization which, for some cell shapes, eventually returns to the original polarization direction (see Fig 5, T = 1000s for L-R). In addition, it would be valuable to study which Rho GTPase circuitry is necessary for geometry sensing. Here, we show, for example, that the minimal model (excluding PI feedback) can only replicate reverse polarization in a rectangle, and not in a circle, triangle or teardrop shape.

The *in silico* results presented in this work are consistent with findings reported in literature. We were able to establish analogous polarization profiles of Rho GTPases to the *in silico* model of Marée et al. [4] which served as a starting point for the *in silico* model in this study – with both Rac and Cdc42 high at the front and low at the rear of the cell and Rho with an opposite trend (S2, S3 Figs in S1 File). Graphs of the spatiotemporal evolution Rho GTPases were also similar but we obtained maximum concentrations about three times higher than Marée et al. [4] as our standard cell's longest axis is three times longer than the one of Marée et al. [4] (50 μm versus 15 μm). Our results also agree with Spill et al. [6] and Jilkine et al. [13], who reported that the peak concentration increases with the size of the cell. Spill et al. [6], also observed that single ellipsoidal (or roundish) cells lost their initial polarization compared to asymmetric cells made up of two ellipsoids connected by a thin neck, mimicking a filopodium. Furthermore, Spill et al. [6], have reported that elongated cells usually polarize along their longest axis, which is the general trend we observed when investigating the role of the cell aspect ratio (Fig 8). As demonstrated by Jiang et al. [14], cells with asymmetric geometries move in a directed manner with the front corresponding to the blunt end while symmetric cells display an unpredictable direction of motion. They further confirm that the asymmetry in geometry alone was responsible for the preferential direction of movement. Despite the occurrence of reverse polarization, our simulations also show that asymmetric shapes, depending on parameter settings, tend to quickly regain their initial polarization direction (towards the blunt end), which may point towards a greater ability to maintain polarization and directed migration. Symmetric shapes take longer and sometimes fail to return to a polarized state (Fig 7A and 7C). Recently, Fink et al. used dumbbell-like micropatterns with various areas, shapes, and orientations to probe cell migration by looking at the dwell times and relative cell occupancy as readouts [33]. They found that on sites of equal area, an asymmetry in the occupancy is induces by anisotropic shapes like triangles in contrast to isotropic shapes like squares and circles, suggesting that the cellular migration between patterns depends on the cell polarization induced by the anisotropic shapes of the micropattern [33]. Interestingly, recent experimental studies have observed wave-like propagation across PI domains [34], a pulsatile nature of Cdc42 activity [35] and directionally reversible Cdc42 activation [36], reminiscent of the "reverse polarization" we predict in this study. Furthermore, Cao et al. [37] have modelled and observed experimentally that the oscillatory behavior of biochemical waves is greatly involved in the switching of migration modes in cells. Particularly, they have shown that the reduction of protrusive forces is involved in this oscillatory behavior. Bolado-Carrancio et al. [38] have shown that ROCK inhibition lead

to the emergence of multiple oscillatory centers, which in turn gave rise to morphological changes; namely cells acquiring large elongated shapes. This finding is consistent with our aspect ratio experiments.

The extensive results summarizing the important role of cell shape, aspect ratio, initial polarization strength and direction, the inactivation rate of Rho GTPases, PI feedback and strength of the negative feedback from Rho to Cdc42 in the emergence of particular Cdc42 patterns should be interpreted in the light of the following assumptions and limitations. Firstly, our computational model only focuses on the biochemical components that interact with the cell shape, thereby neglecting the biomechanical aspects such as membrane tension and cytoskeletal contractility [39]. Secondly, we limit the *in silico* simulations to static, 2D domains, which could be an acceptable translation of 2D culture scenarios but not of dynamic, 3D cellular environments [4, 6]. In the analysis of the results, we also focused on the general trends between symmetric and asymmetric shapes, but it would be interesting to further explore the more detailed differences between circular and square cells for example. Thirdly, we defined reverse polarization as the shifting of the maximal Cdc42 concentration in the opposite direction of the initial polarization direction. In some cases, however, the concentration difference between the maximum and the concentration at 50 μm is very small, resulting in a broad front. As such, it would be interesting to explore the reverse polarization phenomenon further and refine its definition. Fourthly, the observed results are highly dependent on the particular parameter values, some of which are estimated and unknown. As such, future work should focus on accurately measuring the parameter values (and in particular of $d_{Cdc42}$ as it has a substantial effect on reverse polarization) and corroborating the *in silico* results by quantifying the spatiotemporal Rho GTPases activity in migrating cells. Exciting new tools are continuously being developed in this area. For example, fluorescence resonance energy transfer (FRET)-based biosensors of RhoA, Rac1 and Cdc42 have been used to extract their subtle signaling patterns and relate them to edge protrusion and retraction dynamics [40], and migration characteristics such directionality (related to a sharp Cdc42 gradient at the front of the cell) and speed (related to an extended Rac1 gradient) [41]. Other strategies to perturb the activities of Rho GTPases in living cells, including knockdown, knock out, overexpression, microinjection, and optogenetics, are reviewed in Goedhart et al. [42]. In addition the position of the Golgi body relative to the nucleus may also be a useful metric to study cell polarity dynamics [43].

## Conclusion

In this study, we studied the spatiotemporal evolution of Cdc42 for different shape categories (asymmetric and symmetric) to explore the effect of cell shape on cell polarization patterns. We showed that cell shape and aspect ratio highly influence the emergent polarization patterns in addition to the direction and strength of the initial polarization stimulus. The *in silico* results were consistent with previous computational work. Future work should focus on the experimental validation of our *in silico* predictions through, for example, the spatiotemporal tracking of Rho GTPase patterns in migrating cells with FRET probes. In summary, this study shows that cell shape can induce a variety of polarization behaviors.

## Supporting information

**S1 File.**
(DOCX)

## Acknowledgments

We thank Hang Nguyen for her feedback on the text and Figures and Jasia King for independently confirming the simulation results. We are grateful to Leah Edelstein-Keshet for useful discussion and the reviewers for helpful suggestions.

## Author Contributions

**Conceptualization:** A. Vasilevich, S. Vermeulen, J. de Boer, A. Carlier.

**Formal analysis:** K. Eroumé, A. Vasilevich, S. Vermeulen, J. de Boer.

**Funding acquisition:** J. de Boer, A. Carlier.

**Investigation:** K. Eroumé, A. Vasilevich, S. Vermeulen.

**Methodology:** A. Vasilevich, S. Vermeulen, J. de Boer.

**Supervision:** J. de Boer, A. Carlier.

**Visualization:** K. Eroumé.

**Writing – original draft:** K. Eroumé, A. Carlier.

**Writing – review & editing:** K. Eroumé, A. Vasilevich, S. Vermeulen, J. de Boer, A. Carlier.

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
