## [Decision Letter · Decision Letter 0]

8 Jan 2021

PONE-D-20-35237

On the influence of cell shape on dynamic diffusion-reaction polarization patterns

PLOS ONE

Dear Dr. Carlier,

Thank you for submitting your manuscript to PLOS ONE. After careful consideration, we feel that it has merit but does not fully meet PLOS ONE’s publication criteria as it currently stands. Therefore, we invite you to submit a revised version of the manuscript that addresses the points raised during the review process.

Please respond to all comments of the reviewer and modify the manuscript accordingly,

We look forward to receiving your revised manuscript.

Kind regards,

Ivan R. Nabi, Ph.D.

Academic Editor

PLOS ONE

Journal Requirements:

Reviewers' comments:

Reviewer's Responses to Questions

**Comments to the Author**

1. Is the manuscript technically sound, and do the data support the conclusions?

Reviewer #1: Partly

2. Has the statistical analysis been performed appropriately and rigorously? 

Reviewer #1: N/A

3. Have the authors made all data underlying the findings in their manuscript fully available?

Reviewer #1: Yes

4. Is the manuscript presented in an intelligible fashion and written in standard English?

Reviewer #1: Yes

5. Review Comments to the Author

Reviewer #1: Review of Eroumé, K. et al., “On the influence of cell shape on dynamic diffusion-reaction polarization patterns”

In this paper, the authors simulate an existing reaction-diffusion partial differential equation mathematical model of GTPase (Cdc42, Rac, Rho) polarization in several different cellular geometries inspired by the confinement geometries used cell confinement-and-release experiments of Jiang et al. (ref. [14]). The authors seek to study the effect of cell shape (i.e., geometry, aspect ratio, and size) on the formation of GTPase polarization patterns and explore the role of the critical signaling parameters on the polarization process. The authors focus on Cdc42 patterns and find that the maximum Cdc42 activity in the cell propagates towards the cell “rear” in a process that they call “reverse polarization.” The authors systematically vary model parameters to examine how various factors affect polarization. They find that (1) cell shape modulates the Cdc42 activity; (2) that the direction and magnitude of the initial polarization affects the polarization direction in a size and geometry-dependent manner; (3) cell size and aspect ratio affect polarization patterns; and (4) that the “reverse polarization” process is mediated by many factors including geometry and model parameters.

I appreciate the systematic approach used in the paper and that the authors have examined the many facets (geometries, parameter values) through extensive simulations and many detailed supplementary figures. I also commend the authors on using Virtual Cell (a community supported and developed simulation platform). However, I have some questions and comments that should be addressed before publication. Major questions surround the (1) presentation and communication of the results; (2) comparison in Figure 6 and 72; (3) the connection to experimental data from Jiang et al. (ref. [14]); and (4) the relevancy of “reverse polarization” in the system. Minor comments surround the presentation of

Major questions/comments:

(1) Presentation/Communication: The abstract and Figure 1 focuses extensively on the “reverse polarization” concept; however, it seems that “reverse polarization” is more of a model prediction than the focus of the paper. I suggest that Figure 1 be revised to provide an overview of the paper. Perhaps a cartoon or flowchart of sorts that would explain how the model from Maree et al. will be simulated in the geometries from Jiang et al., to determine how the cell shape and cell geometry affect the polarization patterns would better orient the reader to the specific questions, methods, and results of the paper. I did not understand the main point of the paper until I had read the entire paper carefully.

My confusion was further compounded by the choice of the color scheme (that seems to artificially introduce sharp changes; there is much discussion of choosing uniform colormaps that easily convert to greyscale and don’t introduce boundaries) and the choice to not plot the dynamics if the concentration is above a user-picked threshold. It later became clear that the width of the white zone is being used as a proxy for where Cdc42 is highly active and to illustrate the reverse polarization process, but this should be explained initially. I find the plots of x-coordinate of the maximum Cdc42 activity over time to be much clearer in showing the reverse polarization process than the activity profiles.

(2) If the goal is to systematically examine the size or behavior of this reverse polarization zone, then does choosing a threshold for Cdc42 activity confound the results? Here is my argument. Imagine the scenario where the same model is simulated in a circle and a square (as in Figure 5). It appears that the Cdc42 patterns are similar, but the area of the white region in square is larger than the area of the white region in the circle. But since both the areas and the shapes are different, I do not think it is possible to determine whether it is the change in shape or the change in area that affects the polarization process without further simulations to separately examine the effects of shape and area (as in Figure 8). The conclusions on pg. 22 state that:

“the maximal Cdc42 concentration shifted earlier in the opposite direction of initial polarization in the symmetric shapes (circle, square and rectangle) compared to the asymmetric shapes (teardrop, narrow drop, triangle, wide drop) (5s versus 15s respectively, Fig 7A). The maximal Cdc42 concentration also travelled the furthest for the symmetric shapes (Fig 7A) in which it crossed the midline of the cell…”

I do not believe that this comparison is justified since there is more than one factor (shape and size) being altered in these simulations. The essential problem is that the x-coordinate axis is unique for each cell shape shown Figure 7. Moreover, the authors already note that the maximal concentration of Cdc42 depends on the size of the domain (larger domain correlates with more activity) so in a larger cell, the Cdc42 maximal area should be larger. Of course, if the goal is to simply show that “reverse polarization” occurs in these shapes, then Figure 7 achieves this goal, but I do not believe that comparing the “x-coordinate of the maximum Cdc42 activity” and “polarization time” across different shapes provides insight.

(3) Jiang et al. report that narrow teardrops support polarization, but wide teardrops do not: “cells confined as wide drops quickly lost their polarity once released from the pattern” (pg. 977 Jiang et al.). The results in Figure 5 indicate that the wide tear drop also supports a polarized pattern. How should these results be reconciled?

(4) One aspect of the paper that I find very interesting is the prediction/discovery of the “reverse polarization” process, where the Cdc42 maximum propagates towards the rear of the cell. This challenge the possibly over-simplified view taken in many models of cell polarity that Cdc42/Rac should be maximal at the cell front and Rho at the rear and should (as the authors note) be checked experimentally using biosensors. That said, there are many examples of “wave-like” or “propagating” behavior known to occur in cell polarity and migration that could be related to this backwards propagation of Cdc42 activity observed. The authors may wish to expand on their discussion of reverse polarization in the paper. See, for example:

- Cao et al., “Plasticity of cell migration resulting from mechanochemical coupling,” eLife 2019, https://doi.org/10.7554/eLife.48478: shows a wave of backward propagating activity during amoeboid-like migration (Video 2)

- Knoch et al., “Modeling self-organized spatio-temporal patterns of PIP3 and PTEN during spontaneous cell polarization,” Physical Biology 2014, http://dx.doi.org/10.1088/1478-3975/11/4/046002: shows PIP3 waves that propagate toward the cell “rear”.

- Bolado-Carrancio et al., “Periodic propagating waves coordinate RhoGTPase network dynamics at the leading and trailing edges during cell migration,” eLife 2020, https://doi.org/10.7554/eLife.58165: describes periodic propagating waves of GTPase activity during migration.

Minor questions/comments:

(1) Reproducibility: Some additional details on accessing the code in the VCell repository/database would be helpful.

a. It is not obvious to me how to reproduce the results from your paper or to download the code.

b. How were the geometries defined?

c. How was the parameter sensitivity analysis carried out? Details of the parameters are provided in the methods section on pg. 18 but there is no explanation of what was done. Were the parameters all changed at once or one-by-one?

(2) Figure 3 could use word labels to define the shapes since they are discussed in the text in words, e.g., “tear drop”, “wide tear-drop.” Which shapes are considered “symmetric” vs. “asymmetric”? I was confused by these words since the tear-dop shape does have an axis of symmetry.

(3) Figure 6: why are the teardrops in the middle column pointing in the opposite direction (compared to the first and last columns)?

(4) Does the fact that there is a discontinuity in the initial polarization stimulation function explain the jagged lines in some of the figures (e.g., Figure S4). I would expect smooth dynamics for the evolution of the variables in the domains, yet there are very sharp and jagged transitions in shown throughout the paper (in Figure 7, Supporting Figure 19 and 21, etc.). Do these sharp transitions/jagged changes indicate issues with the numerical methods used?

(5) Figure 8B: Could use additional explanation. I still don’t fully understand this figure. Perhaps the authors could walk the reader through interpreting one subpanel of the as an example in the text.

(6) “Exploring the parameter space of reverse polarization:” the detailed quantifications in Figures S11-21 are provided without explanation. Some text preceding Figure 9 should explain how the spatial evolution of the “activation-inactivation” term relates to the Cdc42 dynamics. I think the reader could benefit from an explanation of how Cdc42 is activated when this term is positive, and the spatial extent of Cdc42 activation changes spatially over time.

(7) Text after Figure 9 could use further explanation. “Altering the (membrane) diffusion coefficient resulted in…” could describe how specific changes had distinct effects on the results.

(8) The authors may also wish to connect their work with other experimental papers studying GTPase activity in polarization:

a. Yang HW, Collins SR, Meyer T. Locally excitable Cdc42 signals steer cells during chemotaxis. Nat Cell Biol. 2016;18: 191–201. doi:10.1038/ncb3292

b. O’Neill PR, Kalyanaraman V, Gautam N. Subcellular optogenetic activation of Cdc42 controls local and distal signaling to drive immune cell migration. MBoC. 2016;27: 1442–1450. doi:10.1091/mbc.E15-12-0832

c. Machacek M, Hodgson L, Welch C, Elliott H, Pertz O, Nalbant P, et al. Coordination of Rho GTPase activities during cell protrusion. Nature. 2009;461: 99–103. doi:10.1038/nature08242

Miscellaneous/Optional comments:

(1) Typographical errors:

a. 4th line pg 9: RhoGTPases is missing a space

b. i.e. and e.g. require commas after the second period throughout (“i.e.,”, “e.g.,”);

c. some mathematical equations are not formatted using “math mode” but are left as text (for example on pg. 12 “the parameter f (0 ≤ f ≤ 1) was used” but f is typeset not as an equation. There are more formatting mistakes and inconsistent subscripting in the equations and in tables (for example, D subscript p vs. D subscript P in table 1 and equations 5-6, and on pg 31: a1, dcdc42 should be formatted properly.

d. Reference [27] is missing journal details.

e. Are there references for the supporting material? Ref [3] is referred to in the “2. Minimal Model” section; however, there is no reference list provided.

(2) Eqn. 6: There are two terms with k_PI5K. The second one is a mistake and should be k_PTEN.

(3) pg. 14: “The rate of Cdc42 amplification by Rac is given by alpha” is written backwards. The equations describe how Rac activity is increased from baseline levels I_r to I_r + alpha*Cdc42 therefore alpha describes how Rac is amplified by Cdc42. Similarly, for beta.

(4) Figure 10 caption: “black phase” should be “grey phase”

(5) Fig S1: What does “Evolution” refer to? I think the caption title should be revised “Total concentration of Cdc42 at t = 500 s for different number of mesh elements.”

(6) Typically, “reaction-diffusion” is used by the community to describe these types of PDE models (instead of “diffusion-reaction”). This change is completely optional and left up to the author’s discretion.

6. PLOS authors have the option to publish the peer review history of their article (what does this mean?). If published, this will include your full peer review and any attached files.

Reviewer #1: No

---

## [Author Response · Author response to Decision Letter 0]

8 Feb 2021

Response to Reviewers

We would like to thank the reviewers for their appreciation of our work, including “the systematic approach used in the paper”, “extensive simulations and many detailed supplementary figures” and “the very interesting prediction/discovery of the “reverse polarization” process” as well as for their valuable comments. We have revised the manuscript according to their suggestions, and hope that they will appreciate the quality improvement of the manuscript. We have included a point-by-point reply to their comments and suggestions where appropriate (original reviewers’ comments in italic/bold). All updates in the revised manuscript are presented as track changes. For the reviewers’ convenience, revised sections of the manuscript have been included in the response to the reviewers (between quotes, again with revisions underlined and in italic).

Reviewer #1 

(1) Presentation/Communication: The abstract and Figure 1 focuses extensively on the “reverse polarization” concept; however, it seems that “reverse polarization” is more of a model prediction than the focus of the paper. I suggest that Figure 1 be revised to provide an overview of the paper. Perhaps a cartoon or flowchart of sorts that would explain how the model from Maree et al. will be simulated in the geometries from Jiang et al., to determine how the cell shape and cell geometry affect the polarization patterns would better orient the reader to the specific questions, methods, and results of the paper. I did not understand the main point of the paper until I had read the entire paper carefully.

My confusion was further compounded by the choice of the color scheme (that seems to artificially introduce sharp changes; there is much discussion of choosing uniform colormaps that easily convert to greyscale and don’t introduce boundaries) and the choice to not plot the dynamics if the concentration is above a user-picked threshold. It later became clear that the width of the white zone is being used as a proxy for where Cdc42 is highly active and to illustrate the reverse polarization process, but this should be explained initially. I find the plots of x-coordinate of the maximum Cdc42 activity over time to be much clearer in showing the reverse polarization process than the activity profiles. 

We thank the reviewer for the valuable suggestions regarding the figures in the manuscript although we are not sure we fully understand the reviewer’s comment. We agree that the coloring of figure 1 is confusing, suggesting reverse polarization and as such we have adapted the coloring scheme. Regarding Figures 5-6 for example, although, we agree with the reviewer that the plots of the x-coordinates are more informative, they complicate summarizing all results in a compact figure. As such, we provide the detailed x-coordinate figures in the supplementary material, and have explained the meaning of the white zone, as well as the selection for the plotting threshold in the material and methods.

Manuscript section – Material and methods – 4. In silico analyses: 

“For the parameter sensitivity analysis we varied the PI feedback (f = 0, 0.1, 0.8), the inactivation rate of the Rho GTPases (dcdc42, drac, drho = 0.01, 0.1, 1, 10 s-1), and the strength of the negative feedback of Rho on Cdc42 (a1 = 0.5, 1.25, 2 µM) for the teardrop, and circle cell shapes. We also varied the Cdc42-Rho mutual inhibition coefficient (n = 1, 4, 8), the diffusion coefficients of active (DCdc42,DRac, DRh0 = 0.01, 1 µm2 s-1) and inactive (DCdc42, DRac, DRh0 = 5, 500 µm2 s-1) Rho GTPases. Finally, we varied the diffusion coefficients of all PIs (DPIP1, DPIP, DPIP3 = 5, 500 µm2 s-1). The standard polarization scheme was used. The (threshold) maximum concentrations presented in white in the activity profiles in the results section (from Figure 4 onwards) were determined such that the maximum concentration at different time points for each species across all shapes could be visualized by the same maximum.”

Revised Figure 1:

Figure 1: Summary chart of the investigation of the influence of cell shape on cell polarization gradients. From left to right; schematic overview of cell polarization (top) Unstimulated cell (bottom) Following stimulation (e.g. mechanical or biochemical) of a cell, active Rho GTPases redistribute such that regions of high active Rac and Cdc42 represent the front while the back is associated with regions of a high amount of active Rho as suggested by [4]. We implemented the computational model of Marée et al. [4] on the shapes used by Jiang et al. [14] to investigate the influence of cell shape on polarization, including the effect of polarization direction, polarization strength, size, shape and aspect ratio.

(2) If the goal is to systematically examine the size or behavior of this reverse polarization zone, then does choosing a threshold for Cdc42 activity confound the results? Here is my argument. Imagine the scenario where the same model is simulated in a circle and a square (as in Figure 5). It appears that the Cdc42 patterns are similar, but the area of the white region in square is larger than the area of the white region in the circle. But since both the areas and the shapes are different, I do not think it is possible to determine whether it is the change in shape or the change in area that affects the polarization process without further simulations to separately examine the effects of shape and area (as in Figure 8). The conclusions on pg. 22 state that:

“the maximal Cdc42 concentration shifted earlier in the opposite direction of initial polarization in the symmetric shapes (circle, square and rectangle) compared to the asymmetric shapes (teardrop, narrow drop, triangle, wide drop) (5s versus 15s respectively, Fig 7A). The maximal Cdc42 concentration also travelled the furthest for the symmetric shapes (Fig 7A) in which it crossed the midline of the cell…”

I do not believe that this comparison is justified since there is more than one factor (shape and size) being altered in these simulations. The essential problem is that the x-coordinate axis is unique for each cell shape shown Figure 7. Moreover, the authors already note that the maximal concentration of Cdc42 depends on the size of the domain (larger domain correlates with more activity) so in a larger cell, the Cdc42 maximal area should be larger. Of course, if the goal is to simply show that “reverse polarization” occurs in these shapes, then Figure 7 achieves this goal, but I do not believe that comparing the “x-coordinate of the maximum Cdc42 activity” and “polarization time” across different shapes provides insight.

We agree with the reviewer for raising this valid point. However, we believe that the domain size is most important in the direction of polarization (which we do keep constant in Figure 7). To address the concerns of the reviewer we have phrased our conclusions more carefully and added sentences in the results and the discussion sections to highlight that the influence of shape and area needs to be explored more thoroughly.

Manuscript section – Results

“Both asymmetric and symmetric shapes do not regain their original polarization direction after 150s. It is important to highlight that, although the domain size was constant in the direction of polarization for the L-R and R-L initial polarization experiments, it varied between cell shapes which may influence the above results regarding the shift and timing of the maximal concentration (see also Figure 8).”

Manuscript section - Discussion 

“Despite an initial polarization gradient, the in silico results showed that the maximal Cdc42 concentration could shift in the opposite direction, a phenomenon we called “reverse polarization”. To investigate the “reverse polarization” phenomenon, we explored the activation, inactivation and diffusion terms in more detail. More specifically, for medium to large cells (20 µm–90 µm), the aspect ratio influenced the Cdc42 pattern more than the (a)symmetric cell shape. For a similar aspect ratio, additional in silico analyses indicated that cells shift their maximal Cdc42 concentration at varying degrees depending on their shape, the aspect ratio, the direction and the strength of the initial stimulus, the inactivation rate of the Rho GTPases, the PI feedback and strength of the negative feedback from Rho to Cdc42. Moreover, by varying the inhibition coefficient, the inactivation rate and inhibition of Cdc42 by Rho in a combinatorial fashion, we showed that when the inhibition coefficient was higher than 1 reverse polarization could occur in particular regions of the parameter space (i.e. a1 > 0.5 and dcdc42 > 0.1), although within this parameter space some zones with normal polarization were observed as well. Interestingly, also the direction of the initial stimulus (R-L versus L-R) influenced the resulting Cdc42 patterns in the case of an asymmetrical shape such as a teardrop, i.e. the shift in maximal Cdc42 concentration was more pronounced for the R-L direction than the L-R direction. It is worth pointing out that although we kept the length along the axis of polarization constant across the various shapes for the L-R and R-L initial polarization experiments, the domain size varied with shape which may confound the results regarding the shift and timing of the maximal concentration (see also Figure 8). Future work should focus on designing dedicated experiments to separately examine the effects of cell shape and cell area on cell polarization patterns.”

 (3) Jiang et al. report that narrow teardrops support polarization, but wide teardrops do not: “cells confined as wide drops quickly lost their polarity once released from the pattern” (pg. 977 Jiang et al.). The results in Figure 5 indicate that the wide tear drop also supports a polarized pattern. How should these results be reconciled?

We thank the reviewer for the interesting comment. Indeed, in Figure 5, the wide teardrop supports a polarized pattern, however, at 25 seconds, this shape also displays reverse polarization (as is quantified by Figure 7A). Looking at figure 4D of Jiang et al. we see that the wide teardrop maintains polarization almost 40% of the time. As such, these two results, pointing towards a loss of polarity can be reconciled. 

(4) One aspect of the paper that I find very interesting is the prediction/discovery of the “reverse polarization” process, where the Cdc42 maximum propagates towards the rear of the cell. This challenge the possibly over-simplified view taken in many models of cell polarity that Cdc42/Rac should be maximal at the cell front and Rho at the rear and should (as the authors note) be checked experimentally using biosensors. That said, there are many examples of “wave-like” or “propagating” behavior known to occur in cell polarity and migration that could be related to this backwards propagation of Cdc42 activity observed. The authors may wish to expand on their discussion of reverse polarization in the paper. See, for example:

- Cao et al., “Plasticity of cell migration resulting from mechanochemical coupling,” eLife 2019, https://doi.org/10.7554/eLife.48478: shows a wave of backward propagating activity during amoeboid-like migration (Video 2)

- Knoch et al., “Modeling self-organized spatio-temporal patterns of PIP3 and PTEN during spontaneous cell polarization,” Physical Biology 2014, http://dx.doi.org/10.1088/1478-3975/11/4/046002: shows PIP3 waves that propagate toward the cell “rear”.

- Bolado-Carrancio et al., “Periodic propagating waves coordinate RhoGTPase network dynamics at the leading and trailing edges during cell migration,” eLife 2020, https://doi.org/10.7554/eLife.58165: describes periodic propagating waves of GTPase activity during migration.

We thank the reviewer for the compliment on our work and the relevant literature suggestions, which we have added to the discussion section as detailed below.

Manuscript section - Discussion 

 “The in silico results presented in this work are consistent with findings reported in literature. We were able to establish analogous polarization profiles of Rho GTPases to the in silico model of Marée et al. (2012) which served as a starting point for the in silico model in this study � with both Rac and Cdc42 high at the front and low at the rear of the cell and Rho with an opposite trend (S2–S3 Figs). Graphs of the spatiotemporal evolution Rho GTPases were also similar but we obtained maximum concentrations about three times higher than Marée et al. [4] as our standard cell’s longest axis is three times longer than the one of Marée et al. [4] (50 µm versus 15 µm). Our results also agree with Spill et al. [6] and Jilkine et al. [13], who reported that the peak concentration increases with the size of the cell. Spill et al. [6], also observed that single ellipsoidal (or roundish) cells lost their initial polarization compared to asymmetric cells made up of two ellipsoids connected by a thin neck, mimicking a filopodium. Furthermore, Spill et al. [6], have reported that elongated cells usually polarize along their longest axis, which is the general trend we observed when investigating the role of the cell aspect ratio (Fig 8). As demonstrated by Jiang et al. [14], cells with asymmetric geometries move in a directed manner with the front corresponding to the blunt end while symmetric cells display an unpredictable direction of motion. They further confirm that the asymmetry in geometry alone was responsible for the preferential direction of movement. Despite the occurrence of reverse polarization, our simulations also show that asymmetric shapes, depending on parameter settings, tend to quickly regain their initial polarization direction (towards the blunt end), which may point towards a greater ability to maintain polarization and directed migration. Symmetric shapes take longer and sometimes fail to return to a polarized state (Fig 7A,C). Recently, Fink et al. used dumbbell-like micropatterns with various areas, shapes, and orientations to probe cell migration by looking at the dwell times and relative cell occupancy as readouts [33]. They found that on sites of equal area, an asymmetry in the occupancy is induces by anisotropic shapes like triangles in contrast to isotropic shapes like squares and circles, suggesting that the cellular migration between patterns depends on the cell polarization induced by the anisotropic shapes of the micropattern [33]. Interestingly, recent experimental studies have observed wave-like propagation across PI domains (Knoch et al. (2014)), a pulsatile nature of Cdc42 activity (Wong et al. (2015)) and directionally reversible Cdc42 activation (O’Neill et al. (2016)), reminiscent of the “reverse polarization” we predict in this study. Furthermore, Cao et al. (2019) have modelled and observed experimentally that the oscillatory behavior of biochemical waves is greatly involved in the switching of migration modes in cells. Particularly, they have shown that the reduction of protrusive forces is involved in this oscillatory behavior. Balado-Carancio (2020) have shown that ROCK inhibition lead to the emergence of multiple oscillatory centers, which in turn gave rise to morphological changes; namely cells acquiring large elongated shapes. This finding is consistent with our aspect ratio experiments.”

Minor questions/comments:

(1) Reproducibility: Some additional details on accessing the code in the VCell repository/database would be helpful.

a. It is not obvious to me how to reproduce the results from your paper or to download the code.

We would like to thank the reviewer for this comment. We have added a short description on how to access and use the model in the supplementary material.

Supplementary - Model use and access

We have provided a link to the standard polarization model which is available as a public model in the VCell repository as Kerbai_PLoSone_2021_teardrop_polarization

_extended for the extended model and Kerbai_PLoSone_2021_teardrop_polarization

_minimal for the minimal model under the user name KerbaicBITE. Model codes (extended and minimal) are provided in the supplementary (S28_Models). Details on how to run a model in VCell can be found in the quick start guide on the VCell website, https://vcell.org/support.

b. How were the geometries defined?

We thank the reviewer for this important comment. We have modified the materials and methods as well as the supplementary parts accordingly.

In the Materials and methods – Geometries section we reference the supplementary:

“(Fig 3 and the geometry definition section of the supplementary material)”

Supplementary - Geometry definition

An analytic expression was used to define the symmetric shapes (circle, rectangle and square) in Virtual Cell. This can be specified in the geometry definition menu of the current model. The asymmetric shapes (teardrop, wide drop, narrow drop and triangle) were created by importing images of those shapes into Virtual Cell (see S29). For every shape the various domains, cell, nucleus and extracellular regions were then defined. The major axis length was set at 50 µm and kept constant across the shapes. Each domain was finally mapped to its corresponding region and named accordingly in the structure mapping menu of Virtual Cell. Further details of the geometry definition process can be found in the “quick start guide” or in one of the tutorials at https://vcell.org/support. 

c. How was the parameter sensitivity analysis carried out? Details of the parameters are provided in the methods section on pg. 18 but there is no explanation of what was done. Were the parameters all changed at once or one-by-one?

We agree with the reviewer that the materials and methods lacked this description, and have added it to the materials and methods section. 

Material and methods – In silico analyses

“For the parameter sensitivity analysis we varied the PI feedback (f = 0, 0.1, 0.8), the inactivation rate of the Rho GTPases (dcdc42, drac, drho = 0.01, 0.1, 1, 10 s-1), and the strength of the negative feedback of Rho on Cdc42 (a1 = 0.5, 1.25, 2 µM) for the teardrop, and circle cell shapes. We also varied the Cdc42-Rho mutual inhibition coefficient (n = 1, 4, 8), the diffusion coefficients of active (DCdc42,DRac, DRh0 = 0.01, 1 µm2 s-1) and inactive (DCdc42, DRac, DRh0 = 5, 500 µm2 s-1) Rho GTPases. Finally, we varied the diffusion coefficients of all PIs (DPIP1, DPIP, DPIP3 = 5, 500 µm2 s-1). The standard polarization scheme was used. We first performed a 1D sensitivity analysis by varying the activation, inactivation and diffusion terms one-by-one. We then proceeded with a 2D sensitivity analysis by varying a1 and dCdc42 simultaneously at specific values of n (1, 4, 8).”

 (2) Figure 3 could use word labels to define the shapes since they are discussed in the text in words, e.g., “tear drop”, “wide tear-drop.” Which shapes are considered “symmetric” vs. “asymmetric”? I was confused by these words since the tear-dop shape does have an axis of symmetry.

We thank the reviewer for this comment. We have clarified figure 3 and the materials and methods:

- Revised Figure 3:

- Material and methods - Geometries:

“We distinguished between “symmetric” and “asymmetric” shapes. The symmetric shapes have more than one axis of symmetry (i.e. the circle, square and rectangle) while the asymmetric shapes only have one axis of symmetry (i.e. the teardrop, wide drop, narrow drop and triangle). The cell geometries were defined in the geometry definition menu of Virtual Cell software by using analytical expressions for the symmetric geometries, and by importing images of the asymmetric geometries (Fig 3 and supplementary material).”

(3) Figure 6: why are the teardrops in the middle column pointing in the opposite direction (compared to the first and last columns)?

We would like to apologize and thank the reviewer for spotting this error. The shapes should indeed be reversed, and we have adapted the figure accordingly. 

Revised Figure 6: 

 (4) Does the fact that there is a discontinuity in the initial polarization stimulation function explain the jagged lines in some of the figures (e.g., Figure S4). I would expect smooth dynamics for the evolution of the variables in the domains, yet there are very sharp and jagged transitions in shown throughout the paper (in Figure 7, Supporting Figure 19 and 21, etc.). Do these sharp transitions/jagged changes indicate issues with the numerical methods used?

We thank the reviewer for this remark, which inspired us to check the simulation results thoroughly again. For Figure 7A, the jagged lines are real results, where the maximum sometimes stays at a particular location for a certain time before changing position. Figures 7B, S19 and S21 contained some incorrect and spatial sampling, which we corrected in the final version. We apologize for this error. 

Revised Figure 7:

Revised Figure S19:

Revised Figure S21:

(5) Figure 8B: Could use additional explanation. I still don’t fully understand this figure. Perhaps the authors could walk the reader through interpreting one subpanel of the as an example in the text.

We thank the reviewer for this comment. We have added an additional explanation to clarify the figure: 

“Figure 8B summarizes the results across three cell shapes (i.e. rectangle, teardrop and circle). Every data point combines the results of three shapes along the two main axes, representing the cell size along these axes. The changes in polarization, which are color-coded and explained in the legend, are recorded for each of these three shapes for each axis length. After 1000s, various polarization patterns emerge for all aspect ratios and all shapes (Fig 8A, S7 and S9 Figs). For the teardrop, small cells (lengths ranging from 10 µm to 20 μm) polarized along the new axis as the cell size along this new axis increased (red color in Fig 8B). Further increasing the size of the cell along the new axis (beyond 30 μm), led to reverse polarization along this axis (Fig 8B), defined as shifting of the maximal Cdc42 concentration in the opposite direction of the initial polarization direction. Note the exception to this general trend, i.e., three cells with length 20 µm and widths between 40 µm – 60 μm maintain their initial polarization. Interestingly, medium-size cells (lengths ranging from 30 μm to 60 μm) maintained the direction of the initial polarization pattern, even when the cell size is increased along the new axis (Fig 8B, blue and green colors). Large cells (lengths greater than 60 µm) only show reverse polarization along the initial axis, even when the cell size along the new axis is increased.”

 (6) “Exploring the parameter space of reverse polarization:” the detailed quantifications in Figures S11-21 are provided without explanation. Some text preceding Figure 9 should explain how the spatial evolution of the “activation-inactivation” term relates to the Cdc42 dynamics. I think the reader could benefit from an explanation of how Cdc42 is activated when this term is positive, and the spatial extent of Cdc42 activation changes spatially over time.

We thank the reviewer for this comment. We have added an additional sentence to clarify the text: 

“To explore the dependence of the shifting of the maximal Cdc42 concentration in the direction opposite the initial polarization direction (termed here “reverse polarization”) on the parameter values, we looked in more detail at equation 1, which describes the spatiotemporal evolution of the Rho GTPases. The change in local Cdc42 concentration depends on an activation, inactivation, and diffusion term. We hypothesized that, depending the parameter values, the balance between activation and inactivation is different locally, moving the maximal Cdc42 concentration in the opposite direction of the initial polarization direction. More specifically, the initial polarization is maintained when this difference is positive while reverse polarization occurs when this term is negative. We first altered the parameter values of the activation, inactivation and diffusion terms in a one-at-a-time analysis (spatial results are shown in Fig 9, detailed quantifications can be found in S11–21 Figs).”

(7) Text after Figure 9 could use further explanation. “Altering the (membrane) diffusion coefficient resulted in…” could describe how specific changes had distinct effects on the results.

We are not sure whether we fully understand the comment of the reviewer. The distinct effects are explained in the accompanying text, as indicated below in italic and underlined:

“Altering the (membrane) diffusion coefficient of the active RhoGTPasesRho GTPases resulted in reverse polarization for both the circle and the teardrop (see Fig 9 and S14A, B Fig for detailed quantifications). In contrast, reduced (cytosolic) diffusion of the inactive RhoGTPasesRho GTPases resulted in a larger shift of the maximal Cdc42 concentration in the opposite direction of the initial polarization direction, although the concentration difference between the maximal Cdc42 concentration and the concentration at 50 µm (right-hand side of the cell) was lower than for the standard condition (Fig 9 and S14 C, D Fig).”

(8) The authors may also wish to connect their work with other experimental papers studying GTPase activity in polarization:

a. Yang HW, Collins SR, Meyer T. Locally excitable Cdc42 signals steer cells during chemotaxis. Nat Cell Biol. 2016;18: 191–201. doi:10.1038/ncb3292

b. O’Neill PR, Kalyanaraman V, Gautam N. Subcellular optogenetic activation of Cdc42 controls local and distal signaling to drive immune cell migration. MBoC. 2016;27: 1442–1450. doi:10.1091/mbc.E15-12-0832

c. Machacek M, Hodgson L, Welch C, Elliott H, Pertz O, Nalbant P, et al. Coordination of Rho GTPase activities during cell protrusion. Nature. 2009;461: 99–103. doi:10.1038/nature08242

We thank the reviewer for the interesting literature suggestions, which we have added to the discussion section as detailed below.

Manuscript section - Discussion 

 “The in silico results presented in this work are consistent with findings reported in literature. We were able to establish analogous polarization profiles of Rho GTPases to the in silico model of Marée et al. (2012) which served as a starting point for the in silico model in this study � with both Rac and Cdc42 high at the front and low at the rear of the cell and Rho with an opposite trend (S2–S3 Figs). Graphs of the spatiotemporal evolution Rho GTPases were also similar but we obtained maximum concentrations about three times higher than Marée et al. [4] as our standard cell’s longest axis is three times longer than the one of Marée et al. [4] (50 µm versus 15 µm). Our results also agree with Spill et al. [6] and Jilkine et al. [13], who reported that the peak concentration increases with the size of the cell. Spill et al. [6], also observed that single ellipsoidal (or roundish) cells lost their initial polarization compared to asymmetric cells made up of two ellipsoids connected by a thin neck, mimicking a filopodium. Furthermore, Spill et al. [6], have reported that elongated cells usually polarize along their longest axis, which is the general trend we observed when investigating the role of the cell aspect ratio (Fig 8). As demonstrated by Jiang et al. [14], cells with asymmetric geometries move in a directed manner with the front corresponding to the blunt end while symmetric cells display an unpredictable direction of motion. They further confirm that the asymmetry in geometry alone was responsible for the preferential direction of movement. Despite the occurrence of reverse polarization, our simulations also show that asymmetric shapes, depending on parameter settings, tend to quickly regain their initial polarization direction (towards the blunt end), which may point towards a greater ability to maintain polarization and directed migration. Symmetric shapes take longer and sometimes fail to return to a polarized state (Fig 7A,C). Recently, Fink et al. used dumbbell-like micropatterns with various areas, shapes, and orientations to probe cell migration by looking at the dwell times and relative cell occupancy as readouts [33]. They found that on sites of equal area, an asymmetry in the occupancy is induces by anisotropic shapes like triangles in contrast to isotropic shapes like squares and circles, suggesting that the cellular migration between patterns depends on the cell polarization induced by the anisotropic shapes of the micropattern [33]. Interestingly, recent experimental studies have observed wave-like propagation across PI domains (Knoch et al. (2014)), a pulsatile nature of Cdc42 activity (Wong et al. (2015)) and directionally reversible Cdc42 activation (O’Neill et al. (2016)), reminiscent of the “reverse polarization” we predict in this study. Furthermore, Cao et al. (2019) have modelled and observed experimentally that the oscillatory behavior of biochemical waves is greatly involved in the switching of migration modes in cells. Particularly, they have shown that the reduction of protrusive forces is involved in this oscillatory behavior. Balado-Carancio (2020) have shown that ROCK inhibition lead to the emergence of multiple oscillatory centers, which in turn gave rise to morphological changes; namely cells acquiring large elongated shapes. This finding is consistent with our aspect ratio experiments.”

 (1) Typographical errors:

We thank the reviewer for spotting these typographical errors. We have carefully read through the document and adjusted them according to the suggestions.

---

## [Decision Letter · Decision Letter 1]

24 Feb 2021

On the influence of cell shape on dynamic reaction-diffusion polarization patterns

PONE-D-20-35237R1

Dear Dr. Carlier,

We’re pleased to inform you that your manuscript has been judged scientifically suitable for publication and will be formally accepted for publication once it meets all outstanding technical requirements.

Kind regards,

Ivan R. Nabi, Ph.D.

Academic Editor

PLOS ONE

Additional Editor Comments (optional):

Reviewers' comments:

Reviewer's Responses to Questions

**Comments to the Author**

1. If the authors have adequately addressed your comments raised in a previous round of review and you feel that this manuscript is now acceptable for publication, you may indicate that here to bypass the “Comments to the Author” section, enter your conflict of interest statement in the “Confidential to Editor” section, and submit your "Accept" recommendation.

Reviewer #1: All comments have been addressed

2. Is the manuscript technically sound, and do the data support the conclusions?

Reviewer #1: Yes

3. Has the statistical analysis been performed appropriately and rigorously? 

Reviewer #1: N/A

4. Have the authors made all data underlying the findings in their manuscript fully available?

Reviewer #1: Yes

5. Is the manuscript presented in an intelligible fashion and written in standard English?

Reviewer #1: Yes

6. Review Comments to the Author

Reviewer #1: The authors have addressed the major and minor comments raised in the previous round of review and I feel that the manuscript is now acceptable for publication.

7. PLOS authors have the option to publish the peer review history of their article (what does this mean?). If published, this will include your full peer review and any attached files.

Reviewer #1: No

---

## [Editor Report · Acceptance letter]

2 Mar 2021

PONE-D-20-35237R1 

On the influence of cell shape on dynamic reaction-diffusion polarization patterns 

Dear Dr. Carlier:

I'm pleased to inform you that your manuscript has been deemed suitable for publication in PLOS ONE. Congratulations! Your manuscript is now with our production department. 

Kind regards, 

on behalf of

Dr. Ivan R. Nabi 

Academic Editor

PLOS ONE